# Engineering surface dipoles on mixed conducting oxides with ultra-thin oxide decoration layers

Matthäus Siebenhofer [1,2] ✉, Andreas Nenning [1], Christoph Rameshan[3], Peter Blaha [4], Jürgen Fleig [1] & Markus Kubicek [1] ✉

Improving materials for energy conversion and storage devices is deeply connected with an optimization of their surfaces and surface modification is a promising strategy on the way to enhance modern energy technologies. This study shows that surface modification with ultra-thin oxide layers allows for a systematic tailoring of the surface dipole and the work function of mixed ionic and electronic conducting oxides, and it introduces the ionic potential of surface cations as a readily accessible descriptor for these effects. The combination of X-ray photoelectron spectroscopy (XPS) and density functional theory (DFT) illustrates that basic oxides with a lower ionic potential than the host material induce a positive surface charge and reduce the work function of the host material and vice versa. As a proof of concept that this strategy is widely applicable to tailor surface properties, we examined the effect of ultra-thin decoration layers on the oxygen exchange kinetics of pristine mixed conducting oxide thin films in very clean conditions by means of in-situ impedance spectroscopy during pulsed laser deposition (i-PLD). The study shows that basic decorations with a reduced surface work function lead to a substantial acceleration of the oxygen exchange on the surfaces of diverse materials.

Optimizing the surface properties of mixed ionic and electronic conducting (MIEC) oxides is a critical research objective within the field of materials for energy storage and conversion technologies. Mass and charge transport across solid/solid, solid/liquid and solid/gas interfaces are fundamental processes in numerous devices such as fuel cells, electrolysis cells and batteries[1–3]. Enhancing the kinetics of these reactions is therefore fundamental for the development of future technologies and to reach decarbonization and climate goals[4]. Alongside tailoring the morphology and composition of materials, surface modification has emerged as an effective strategy to optimize reaction kinetics and device performance[5–7]. However, despite their importance for energy applications, many optimization strategies are still empirically based, and a comprehensive understanding of the

fundamental principles underlying surface modifications remains elusive.

Several pioneering studies have demonstrated that surface modification with sub-nanometer quantities of various oxides can dramatically alter the fundamental properties of the host material's surface, such as oxygen exchange kinetics or interfacial stability[8–11]. These effects have since been associated with changes in vacancy formation energies, surface band bending and the stabilizing influences of the surface layer. In an effort towards a systematic understanding of the impact of oxidic surface modifications, Nicollet et al. introduced the concept of the Smith acidity of the surface oxide as a descriptor for the magnitude of the induced effects[9,12]. Building upon these findings, we have previously demonstrated that acidic adsorbates on the surface

[1]Institute of Chemical Technologies and Analytics, TU Wien, Vienna, Austria. [2]Department of Nuclear Science and Engineering, Massachusetts Institute of Technology, Cambridge, MA, USA. [3]Chair of Physical Chemistry, Montanuniversität Leoben, Leoben, Austria. [4]Institute of Materials Chemistry, TU Wien, Vienna, Austria. ✉e-mail: msieben@mit.edu; markus.kubicek@tuwien.ac.at

of MIEC oxides induce charge transfer from the host material's surface to the adsorbates, consequently forming a surface dipole[13].

In this study, we generalize the concept of induced charge redistribution and dipole formation to ultra-thin oxide decoration layers. We probed the effects of SrO, a highly basic oxide (-9.4 on the Smith acidity scale with a similar basicity as $Li_2O$ at -9.2), and $SnO_2$, a strongly acidic oxide (2.2 on the Smith acidity scale, being slightly less acidic than $WO_3$ at 4.7), on the work function and the electronic structure of $La_{0.6}Sr_{0.4}CoO_{3-\delta}$ (LSC) and $Pr_{0.1}Ce_{0.9}O_{2-\delta}$ (PCO), mixed conducting representatives of two fundamentally different material classes, perovskite and fluorite oxides. Employing both experimental (X-ray photoelectron spectroscopy, XPS) and computational (density functional theory, DFT) methodologies, we showed that basic and acidic oxide decorations induce opposing shifts in the work function of the host oxide. This confirms systematic charge redistribution and dipole formation processes at the surface. Investigating one example of possible applications for this phenomenon, we explored the impact of SrO and $SnO_2$ decoration on the oxygen exchange reaction (OER) on various solid oxide fuel cell (SOFC) cathode materials' surfaces by in-situ impedance spectroscopy during pulsed laser deposition (i-PLD). Our experiments verified that basic SrO decoration enhances and acidic $SnO_2$ decoration deteriorates the OER kinetics across all host materials. Linking this effect to computational results, we propose that the ionic potential of surface cations serves as a readily accessible and suitable descriptor to predict surface properties such as the oxygen exchange activity. From a mechanistic point of view, our results suggest that low work function configurations energetically favor the adsorption of molecular oxygen on the surface. Ultimately, the targeted engineering of surface or interface dipoles might prove critical for the improvement of multiple technologies and devices, including but not limited to fuel cells, batteries, permeation membranes or semiconductor interfaces.

## Results and discussion

### Tuning the work function with ultra-thin oxide layers

To examine the effects of decorations on the properties of the host material, particularly regarding the emergence of surface dipoles, we conducted XPS studies on dense LSC and PCO thin films, both in their pristine state and after decoration. While surface infiltration of porous PCO has already been shown to systematically modulate the work function[9], the results presented here validate this concept across structurally and chemically different materials. As shown in Fig. 1a), the work function of both perovskite LSC and fluorite PCO decreases upon decoration with SrO, and increases upon decoration with $SnO_2$. Differences between LSC and PCO are in good agreement with the surface ionic potential of the host material (see below). From an electrostatic point of view, this implies that basic decorations are positively charged relative to the original surface and vice versa. While work function measurements on LSC were performed at 450 °C and an equivalent $p(O_2)$ of $1 \cdot 10^{-6}$ mbar (see Methodology section), for PCO, the results presented here combine work function measurements on pristine PCO thin films at 600 °C and $8 \cdot 10^{-6}$ mbar with surface potential change measurements via gas phase peak shifts at 600 °C and 1 mbar $O_2$ from one of our previous works[14], utilizing two equivalent ways to assess the qualitative changes of surface dipoles. The decoration layer thickness amounted to 1.0-1.5 nominal monolayers, measured prior to deposition by a quartz balance. Regarding the surface morphology, low energy ion scattering (LEIS) measurements and atomic force microscopy images show very high surface coverage for PCO and LSC thin films decorated with SrO and $SnO_2$[14] as well as no signs of island or particle growth (see supplementary note 8). We therefore assume a flat growth mode of the decoration layer which ultimately leads to a homogenous coverage. Computational results as well as i-PLD experiments support this assumption (see below). While XPS measurements probe a certain sample volume and cannot distinguish between island growth and full layer coverage, LEIS truly probes the outermost surface chemistry and is thus an invaluable tool to investigate modified surfaces.

While previous studies have used the Smith acidity of the decoration as a descriptor to predict these changes[9,13,14], we introduce the ionic potential of surface cations as an alternative and more universal metric to describe surface acidity. Based on the definition by Cartledge[15], we define the surface ionic potential $\phi$ as the ratio of ionic charge $q$ and ionic radius $r$, averaged over the surface cation stoichiometry (with cation fractions $x_i$)

$$\phi = \sum_{i=1}^{n} x_i \frac{q_i}{r_i}. \qquad (1)$$

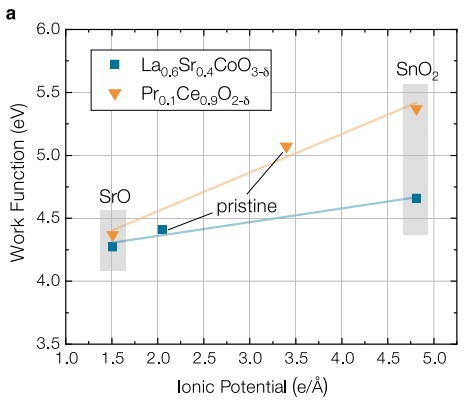
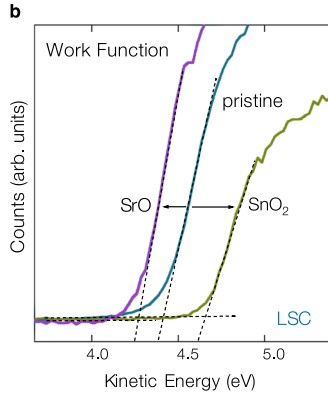
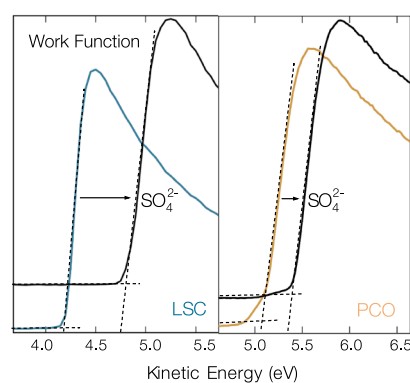

**Fig. 1 | XPS studies of the work function regions of modified $La_{0.6}Sr_{0.4}CoO_{3-\delta}$ (LSC) and $Pr_{0.1}Ce_{0.9}O_{2-\delta}$ (PCO) surfaces. a** Work function changes evaluated by XPS on pristine and decorated (decorations indicated by the gray boxes) $La_{0.6}Sr_{0.4}CoO_{3-\delta}$ (LSC) and $Pr_{0.1}Ce_{0.9}O_{2-\delta}$ (PCO) thin film surfaces in different conditions vs. the estimated ionic potential of the surface. For PCO, surface potential changes at 600 °C and 1 mbar $O_2$, measured in a previous study[14], were added to the work function of a pristine PCO thin film. The weighted ionic potential of pristine LSC was estimated from LEIS measurements of its surface composition[13], for PCO, the nominal stoichiometry (90 % Ce and 10 % Pr) was assumed. **b** XPS measurements of the low kinetic energy cutoff region for pristine, SrO decorated and $SnO_2$ decorated 50 nm LSC thin films at 450 °C and an equivalent $p(O_2)$ of $1 \cdot 10^{-6}$ mbar set via an Fe/FeO reservoir. The dashed lines indicate the fit for work function evaluations. **c** XPS measurements of the low kinetic energy cutoff for pristine and sulfate covered (after exposure to 1 mbar $O_2$) LSC and PCO thin films at 600 °C and $8 \cdot 10^{-6}$ mbar. The dashed lines indicate the fit for work function evaluations. For PCO, a substantially smaller sulfate coverage was estimated from XPS results.

The ionic potential of an ion serves as a measure of the extent to which an ion attracts opposite charge and exhibits strong correlations with both the ionic electronegativity, the ionic character of metal-oxygen bonds in binary oxides and their Smith acidity ($R^2 > 0.7$, see supplementary note 1 for a detailed discussion)[15–17]. High ionic potential ions are usually small and highly charged (such as $Sn^{4+}$), implying high electronegativity, an acidic character of the corresponding oxides and a tendency to form bonds with a stronger covalent character[16]. In contrast, low ionic potential ions exhibit large radii and low charges (such as $Sr^{2+}$), their electronegativity is low, their oxides exhibit a basic character and they tend to form bonds with a stronger ionic character. For this study, we used ionic (crystal) radii from Shannon[18] and the formal charge of the cation in an oxide (e.g. 2 for $Sr^{2+}$ in SrO). The average value of pristine LSC was estimated from LEIS measurements[13], for PCO the nominal stoichiometry (90 % Ce and 10 % Pr) was assumed. It is worth mentioning, that averaging over the surface cation stoichiometry is only an estimate allowing for the comparison of different surface chemistries, and does not account for the oxygen vacancy concentration at the surface as a function of temperature and oxygen partial pressure.

We also probed the effect of acidic adsorbates on the work function of LSC and PCO. Upon exposure of pristine surfaces to considerable gas phase pressures ($> 10^{-3}$ mbar), a new oxygen species appears at 532 eV, alongside a weak corresponding S species, that is caused by the inherent impurity content in measurement atmospheres despite using nominally high purity gases[19]. Consistent with earlier studies, where $SO_4^{2-}$ adsorbates induced a significant increase in the work function of LSC[13], this phenomenon could be reproduced for PCO, albeit with a less pronounced increase. This is in line with the quantification of $SO_4^{2-}$ adsorbates from XPS data, revealing higher concentrations on LSC surfaces (55 % coverage on LSC vs. 25 % coverage on PCO), indicating the strong affinity of the basic LSC surface for acidic adsorbates.

## Dipole formation in heterojunctions with ultra-thin oxide layers

To unravel the underlying atomic scale processes induced by these surface decorations, we performed ab-initio calculations of mixed conducting oxides coated with SrO and $SnO_2$ monolayers as well as with $SO_3$ adsorbates. As a proof of concept, we selected geometric configurations for SrO and $SnO_2$ layers on LSC and PCO based on appropriate bond distances and orientation with regard to the host

lattice as a starting point for structural relaxation (see Fig. 2, for more details refer to Supplementary note 9). SrO-terminated LSC and $SnO_2$-terminated PCO show no significant tendency towards oxygen vacancy or peroxide formation, i.e. removing a neutral oxygen atom or adding a second oxygen atom to a surface oxygen (and thus forming a peroxide) does not lead to energy gains - indicating that the proposed configuration likely reflects an energetically favorable structure. However, for SrO-terminated PCO, peroxide formation is preferable, up to an $SrO_2$ termination (indicating that the initially chosen configuration for SrO-terminated PCO is likely not stable). Here, it is noteworthy that during some XPS experiments on LSC, an intriguing new oxygen species appeared, which might be the first indication of the presence of a peroxide species, but this requires further investigation (see supplementary note 2 for a detailed discussion). For $SnO_2$-terminated LSC, the formation of one surface vacancy is energetically favorable. At this point, we want to emphasize that, similarly to our computational models, also the actual stoichiometry of the deposited decoration layer is unclear (and hardly accessible for analytic tools due to the tiny sample volume). While $SnO_2$ decoration leads to a strong decrease of oxygen exchange kinetics, as would be predicted by its Smith acidity, we cannot exclude the possibility of mixed oxidation states or deviation from ideal oxygen stoichiometry. Based on these idealized models, we evaluated the work function of all structures by calculating the energy difference between the Fermi level and the Coulomb potential at the midpoint of the vacuum space. The work function was then correlated with the ionic potential of surface cations (Fig. 3a).

Ab-initio calculations reveal a strong correlation between the work function and the ionic potential, corroborating the experimental observations from XPS measurements. Oxides with low ionic potential, such as SrO – which are classified as strongly basic oxides under Smith's acidity definition – decrease the work function of the host oxide. Conversely, high ionic potential modifications such as $SnO_2$ and $SO_3$, both considered acidic according to their Smith acidity, significantly increase the host material's work function. Deviations in the absolute work function values between experiment and calculation (e.g. 3.7 eV calc. vs. 4.2 - 4.4 eV exp. for pristine LSC) could stem from multiple sources, including different surface reconstructions, peroxide formation or surface vacancy concentrations at elevated temperatures and higher oxygen partial pressures. In general, surface vacancies lead to reduced work functions and $O_2$ adsorbates or

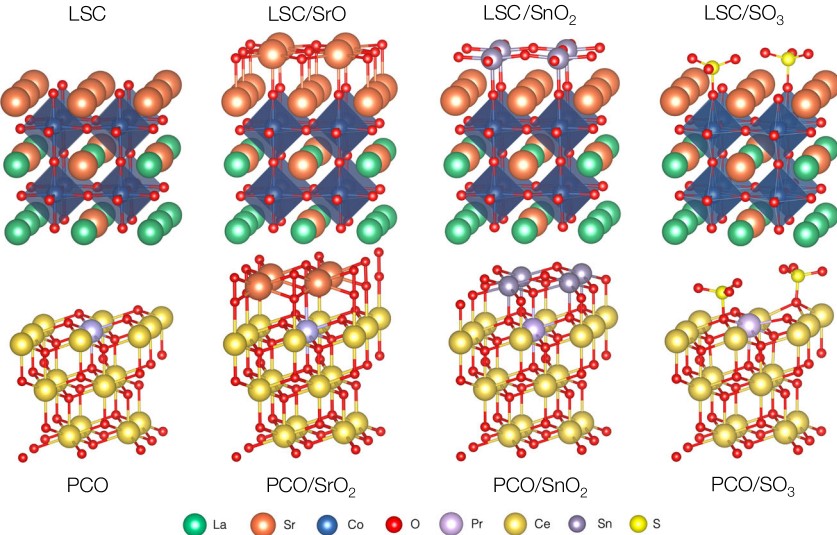

**Fig. 2 | DFT Structures of pristine and decorated La$_{0.6}$Sr$_{0.4}$CoO$_{3-\delta}$ (LSC) and Pr$_{0.1}$Ce$_{0.9}$O$_{2-\delta}$ (PCO).** Decorated/modified mixed conducting oxide structures (La$_{0.6}$Sr$_{0.4}$CoO$_{3-\delta}$ (LSC) and Pr$_{0.1}$Ce$_{0.9}$O$_{2-\delta}$ (PCO)) used for computational

investigations. The figure displays the top halves of the structures, the slabs are completed with adding the inverse structures at the bottom. The whole structures with fractional coordinates are shown in the Supplementary note 9.

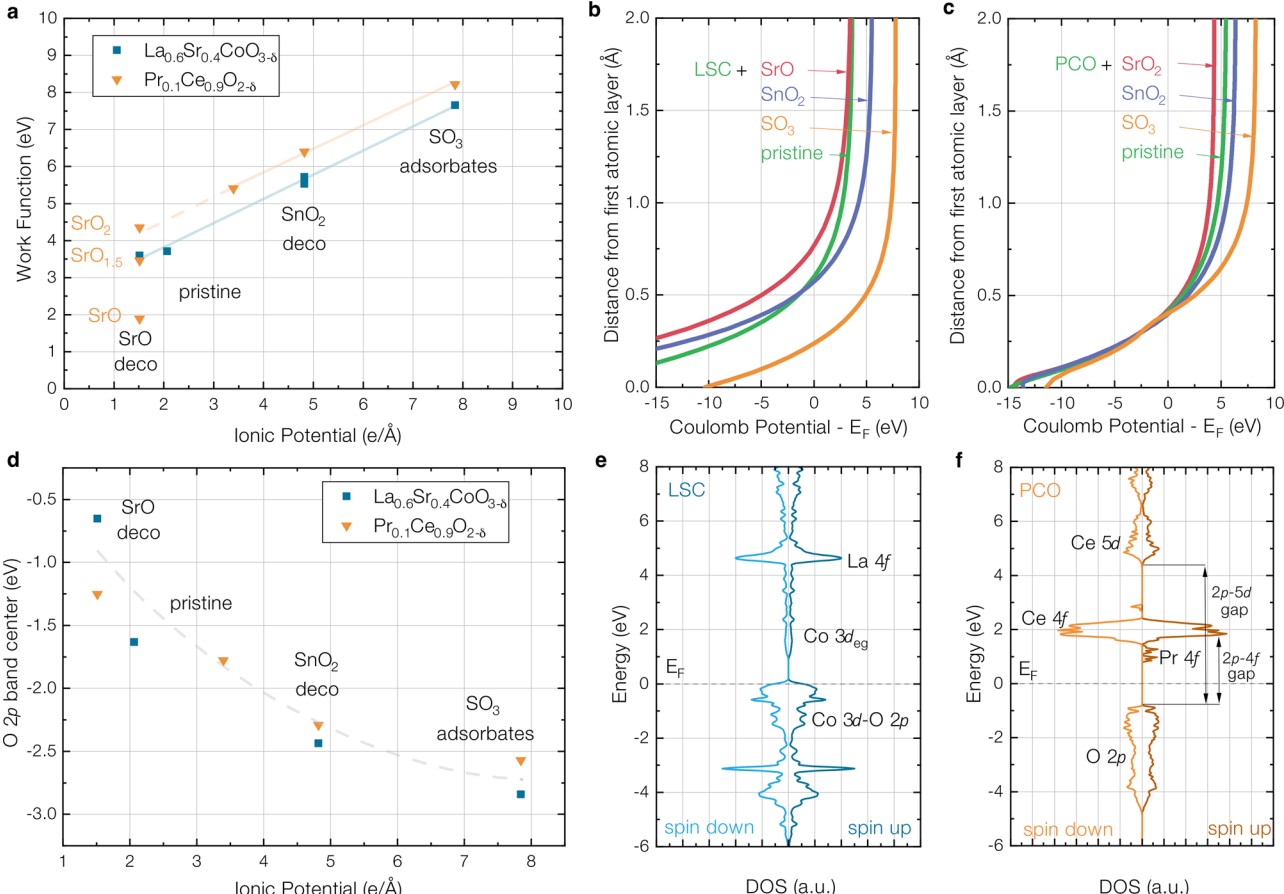

**Fig. 3 | Results of DFT investigations of the work function, the coulomb potential, the surface O 2p band center and the total densities of state of modified La$_{0.6}$Sr$_{0.4}$CoO$_{3-\delta}$ (LSC) and Pr$_{0.1}$Ce$_{0.9}$O$_{2-\delta}$ (PCO). a** Ab-initio work functions of LSC and PCO surfaces, pure or modified with SrO, SnO$_2$ and SO$_3$. In the cluster of values for SnO$_2$-decorated LSC (5.72 eV), also the work functions of SnO$_{1.75}$ (5.55 eV) and SnO (5.53 eV) decorations are shown additionally. For SrO-decorated PCO, the work functions of SrO$_{1.5}$ and SrO$_2$ decorations are also shown. **b** Coulomb potential above the first atomic layer of modified LSC surfaces. **c** Coulomb potential above the first atomic layer of modified PCO surfaces. **d** O 2p band center of surface oxygen vs. the ionic potential for different surface modifications of PCO and LSC. **e** Total density of states for pure LSC. **f** Total density of states for pure PCO with the Fermi level (E$_F$) defined as zero energy.

peroxides to an increased work function, a systematic investigation would, however, require extensive structural variation and goes beyond the scope of this study. The correlation shown above is also displayed and discussed in supplementary note 3 with ionic potentials resulting from ionic radii and ionic charges extracted from a Bader charge analysis[20]. The analysis yields similar trends, emphasizing the fact that a simple estimation of ionic potentials from formal charges and literature values is able to qualitatively predict work function changes on complex oxide surfaces correctly.

This systematic alteration of the work function is linked to surface dipoles that form when MIEC oxides are decorated with ultra-thin oxide layers and which lead to different Coulomb potentials in the vacuum above the surface, as shown in Fig. 3b and c. It is worth mentioning that the Coulomb potential in the center of the slab is independent of the decoration. For a deeper understanding of these dipole formation processes, we analyzed the relaxed structures from ab-initio calculations in greater detail. Depending on the decorating oxide, surface dipoles form due to two primary mechanisms: i) charge transfer between the decoration/adsorbate and the host material's surface (e.g. for SrO decoration on LSC and PCO) and ii) geometric dipole formation via atomic displacement within the decoration/adsorbate layer (e.g. for SnO$_2$ decoration on LSC). Both mechanisms contribute to the buildup of electric fields on the surface, consequently leading to changes in the work function. At this point, it is important to emphasize, that this analysis was done for idealized, stoichiometric monolayers, showcasing fundamental principles of surface dipole formation. As we deal with mixed ionic and electronic conducting materials at high temperatures and in contact with the gas phase, real interfaces likely exhibit higher complexity.

From an electrochemical point of view, dipole emergence is caused by different electrochemical potentials of electrons in the decoration and the material. Specifically, in the exemplary case of a negative surface charge, the electrochemical potential of the electron in the host material initially lies higher than for the decoration. For mixed conductors, depending on a balance between changed defect formation energies (which are also encoded in the electronic structure, e.g. via the O-2p band center[21]) and electrostatic potential buildup, complicated space charge zones might form at high temperatures, entailing altered oxygen vacancy concentrations, nonstoichiometric decoration layers, and complicated reconstructions. Also intermixing between decoration and host material cannot be excluded after longer periods at high temperature. Partly, this complexity is already indicated by our analysis of varying oxygen content in decorations shown in Fig. 3a), where changing oxygen stoichiometry also leads to changes in the work function (examples of the interplay of stoichiometry and surface dipole have also been previously discussed in literature[22]). In addition, the formation of adsorbates on a decoration in contact with a gas phase will also affect the surface dipole as well as the space charge layer and induce further charge transfer. An extended discussion of the

electrochemical potential landscape for the example of SrO decoration on LSC is given in supplementary note 4.

Substantiating this complex picture, DFT calculations also revealed considerable changes of the O-2$p$ band center of the surface modification with its ionic potential (Fig. 3d), with basic decorations inducing an upward shift and vice versa. The O-2$p$ band center, defined as the distance between the Fermi level and the centroid of the O-2$p$ band has recently been in the focus of research as a means to predict the catalytic activity of complex oxides, suggesting that oxides with a shallow O-2$p$ band center also exhibit fast reaction kinetics, e.g. for oxygen surface exchange[23,24].

At this point, we also want to link computational results to the real surface morphology. Based on our surface analysis with LEIS and AFM, decoration layers grow very flat and do not form islands or particles, but possibly also do not cover the surface fully (see Supplementary note 8). Comparing experimental work function measurements with DFT predictions, we observed that the experimentally found trend is indeed weaker than predicted by DFT (especially for LSC, which seems to exhibit a higher fraction of free surface). As XPS work function measurements take an area-weighted average over the work function in a certain spot size[25] and thus might also record signal from small amounts of undecorated surface, this is in good agreement with the surface morphologies and chemistries of our samples.

To comprehensively investigate the evolution of the surface defect chemistry beyond the results presented here, an exact understanding of atomic reconstructions at the surface is necessary. Building upon the findings of this study, a combined approach of high resolution imaging and computational methods could fully uncover the processes that occur during the formation of mixed conducting oxide heterojunctions, thereby pinpointing optimal strategies for tailoring surfaces with specific properties. This could pave the way for a knowledge-driven optimization of various reactions and devices, such as fuel and electrolysis cells, sensors, and batteries. To demonstrate the broad applicability of this concept to surface reactions, we explored the effect of acidic and basic oxides on the OER kinetics of several mixed conducting oxide surfaces at high temperatures.

## Oxygen exchange kinetics modified by surface decorations

To probe the applicability of our theory, we investigated the impact of ultrathin decoration layers on the OER kinetics of chemically and structurally different MIEC oxides. Since exchange kinetics are very prone to impurities, especially sulfur[19], they were measured on pristine surfaces by i-PLD during surface decoration, a technique whose analytic strength was proven in several previous studies[14,19,26,27]. Here, we applied basic SrO and acidic $SnO_2$ decorations to dense $La_{0.6}Sr_{0.4}CoO_{3-\delta}$ (LSC), $La_{0.6}Sr_{0.4}FeO_{3-\delta}$ (LSF), $SrTi_{0.3}Fe_{0.7}O_{3-\delta}$ (STF) and $Pr_{0.1}Ce_{0.9}O_{2-\delta}$ (PCO) thin films and tracked their impact by means of i-PLD (see Fig. 4).

The results of the measurements confirmed what was previously shown for PCO[9,14]. Surface decoration with monolayer amounts of more basic oxides accelerates the OER kinetics on the surfaces of all probed MIEC oxides. Surface decoration with monolayer amounts of more acidic oxides greatly inhibits the oxygen exchange on all MIEC oxide surfaces. In the supporting information (S.I.6), we provide details about the impedance analysis of decorated LSC and PCO and about the i-PLD process. In supplementary note 7, we also provide additional details about the decoration with larger amounts of SrO on LSC and PCO, showing that the optimal kinetics are reached at nominally exactly 1 monolayer decoration thickness, consistent with a very high coverage of the surface by the decoration during i-PLD.

Given that the exact mechanism of the OER is still not clarified, and many different, partially conflicting theories have been brought forward[26,28–31], it is not possible to present a fully comprehensive theory of the effects of decorations on the OER kinetics as of yet. We will, however, discuss the experiments in light of previous results and provide a hypothesis that might reconcile different interpretations. Two important points must be considered in this discussion:

- According to earlier studies, decorations do not alter the OER mechanism but accelerate the reaction rate of the equilibrium reaction, encompassing both oxygen reduction and oxygen evolution[9,14,32]. Possibilities to alter the equilibrium reaction rate include lowering the kinetic barrier of the rate limiting step, changing participating defect concentrations and altering reaction step energetics, thereby providing higher reactant concentrations for the rate limiting step[26]. Previous studies have further shown that decorations do not alter the apparent activation energy of the OER systematically[9,14,26,32–35]. This activation energy is a complex convolution of contributions from equilibrium constants, defect concentrations and the kinetic barrier of the rate limiting step (and possibly even potential-dependent terms)[26]. It is therefore unlikely that a pure decrease of the kinetic barrier of the rate limiting step is the only cause for increased reaction rates.

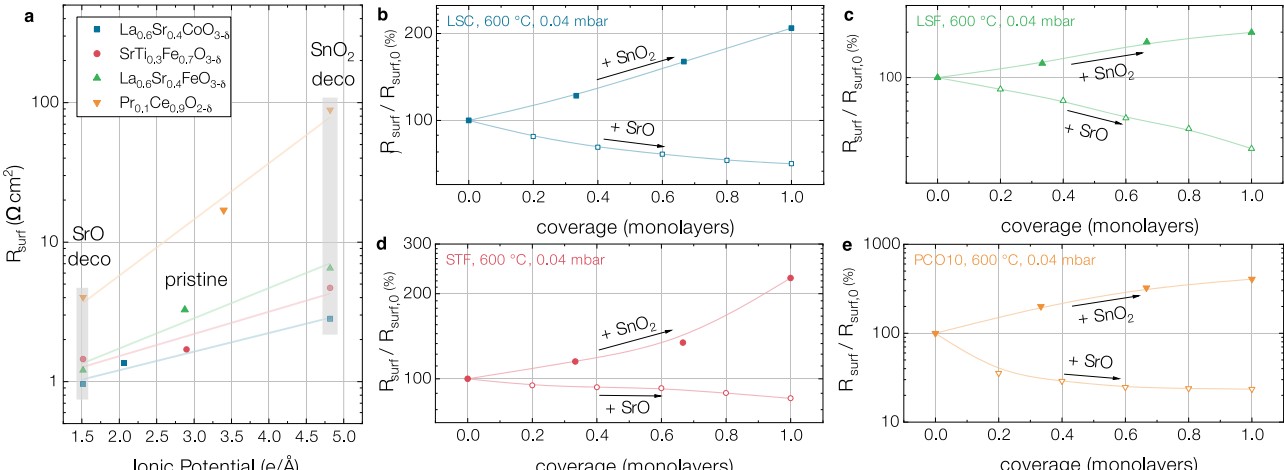

**Fig. 4 | i-PLD measurements of the equilibrium surface exchange resistance upon (sub-)monolayer decorations of mixed conductors with binary oxides. a** Surface exchange resistance of $La_{0.6}Sr_{0.4}CoO_{3-\delta}$ (LSC), $SrTi_{0.3}Fe_{0.7}O_{3-\delta}$ (STF), $La_{0.6}Sr_{0.4}FeO_{3-\delta}$ (LSF) and $Pr_{0.1}Ce_{0.9}O_{2-\delta}$ (PCO) upon surface modification with one nominal monolayer of SrO and $SnO_2$ (indicated by the gray boxes) plotted against the suggested ionic potentials of the compound and the decorations. **b** Stepwise decoration of LSC with SrO and $SnO_2$ and the corresponding change of the surface exchange resistance. **c** Stepwise decoration of LSF. **d** Stepwise decoration of STF. **e** Stepwise decoration of PCO.

- Previous studies of surface decorations have presented conflicting interpretations regarding their working mechanism. While originally, the main effect of surface decorations was attributed to changes in the free surface electron concentration[9], corresponding changes in XPS have not yet been found[9,14], and corresponding conductivity changes could not be reproduced on thin film samples[14]. Instead, adsorption thermodynamics have been discussed as the main reason for the observed changes in OER kinetics[12] and the results of this study support this theory.

At the core of our hypothesis, surface decorations affect the energy levels of adsorbed reaction intermediates, such as peroxo or superoxo species adsorbed at the surface that are likely dissociating during the oxygen incorporation reaction. At this point it is noteworthy that it is unlikely to observe such molecular oxygen adsorbates directly during near ambient pressure (NAP)-XPS, due to their low concentration and because their signal would be masked by higher concentrated bulk oxygen and $SO_4^{2-}$ adsorbates (which are unavoidable at these conditions). However, as shown in Supplementary note 2, it might indeed be possible to provide evidence for these intermediates with sophisticated XPS measurements.

Here, we suggest that low ionic potential (basic) decorations favor the donation of charge to adsorbed oxygen molecules (as has previously been confirmed for metal surfaces and Na coadsorption[36]) and vice versa for high ionic potential (acidic) decorations. This concept is an alternative view on the original argument of electron concentration changes, however, upon basic decoration, charge does not necessarily accumulate in PCO but is rather localized on oxygen adsorbates. The process is schematically shown in Fig. 5d) and leads to different energetics (and concentrations) of charged $O_2$ adsorbates on the decorated surfaces. This is further supported by XPS measurements on SrO-decorated LSC under anodic polarization, where an increased oxygen potential, and hence an increased concentration of surface oxygen species, lead to a work function increase, indicating a more negative surface charge. It is important to note that the band bending picture in this figure is strongly simplified and only illustrates surface dipole formation. In reality, particularly at high temperatures, we expect charge carrier concentrations in the decoration and the surface to evolve according to a complicated interplay of electrostatic and structural effects. This is again supported by the fact that previous studies were generally not able to find corresponding oxidation state changes changes in XPS.

Computational results further support this hypothesis (see Fig. 5a–c): the energetic positioning of the $2p$ orbital of the upper O atom of an $O_2$ molecule on a surface oxygen site gradually shifts towards lower energies (with regard to $E_F$) the more basic the surface and the lower the work function becomes. This leads to a higher concentration of molecular adsorbates in equilibrium conditions at high temperatures, which accelerates the oxygen incorporation for basic decorations and vice versa for acidic decorations (we suspect similar processes to be responsible for the acceleration of the oxygen release reaction as well). We also expect that the higher partial charge on molecular oxygen favors dissociation as the O-O bond length increases. As the second main effect, we suggest that defect concentrations on the surface will be substantially altered during decoration. This is also supported by the shifting O $2p$ band center, which suggests a changing reducibility of the surface[23]. Quantitative estimates, how decorations affect defect concentrations at high temperatures are, however, rather speculative and require a more detailed investigation, possibly including ab-initio thermodynamic approaches. The here presented concept has important implications for studies on the OER and poses a further step towards a comprehensive understanding of degradation processes and optimization strategies for solid oxide cell materials.

Summarizing our study, we used a combination of experimental (XPS, i-PLD) and computational (DFT) techniques to investigate the effect of ultrathin oxide layers on the surface of mixed ionic and electronic conducting host materials. Both approaches confirm that the work function of chemically and structurally different materials can be systematically modified by decoration with different oxides, based on their acidity or basicity. The ionic potential of surface cations has further shown to be an easily accessible descriptor of these effects and can be used for a qualitative initial assessment of the effects of surface modifications and their acidity. To demonstrate the power of this concept, we investigated the oxygen exchange kinetics of four different mixed conducting oxides decorated with low ionic potential (basic) and high ionic potential (acidic) oxides. For all probed combinations, the oxygen exchange kinetics were accelerated with lower ionic potential decorations and therefore by decreasing the work function. This effect might be related to a modification of the energetics of $O_2$ adsorbates and DFT-calculations support this hypothesis.

The presented results emphasize that surface dipoles can be engineered by a targeted surface decoration on mixed conducting surfaces and this concept might pose a strategy to optimize a variety of material properties for numerous applications. Conceptual proof has been given by showing the general applicability of surface decorations to tailor the oxygen exchange kinetics. Continuing with closely related

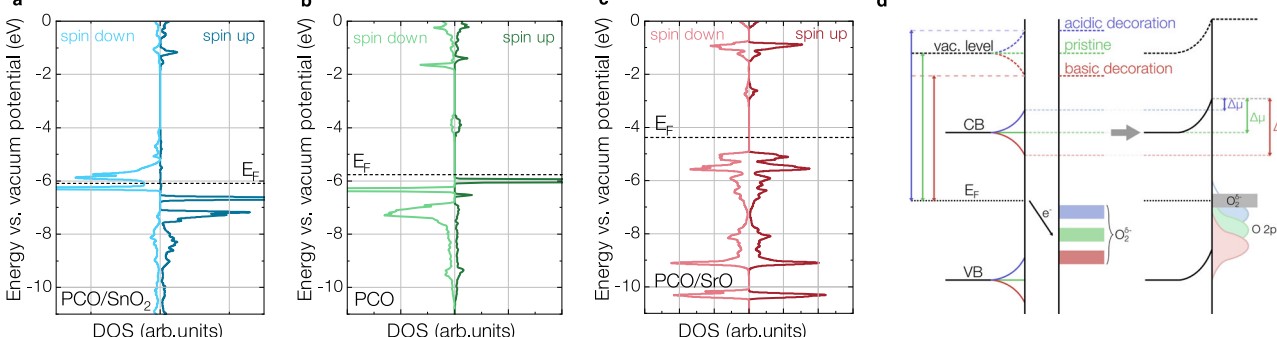

**Fig. 5 | Densities of State of oxygen molecules adsorbed on decorated $Pr_{0.1}Ce_{0.9}O_{2-\delta}$ (PCO) surfaces and schematic of charge transfer upon decoration and adsorption. a** O-pDOS of the upper oxygen atom of an adsorbed $O_2$ molecule on $SnO_2$ decorated $Pr_{0.1}Ce_{0.9}O_{2-\delta}$ (PCO) with the Fermi level ($E_F$) defined as zero energy. **b** O-pDOS for the upper oxygen atom of an adsorbed $O_2$ molecule on undecorated PCO. **c** O-pDOS for the upper oxygen atom of a peroxide unit on $SrO_2$ decorated PCO. **d** Simplified schematic visualization of charge transfer towards $O_2$ adsorbates upon acidic and basic decoration. Acidic and basic decoration lead to a surface dipole, visualized as a band bending in the left part of the diagram (CB is the conduction band edge and VB is the valence band edge). The energetic positioning of O $2p$ orbitals of adsorbed $O_2$ molecules relative to the Fermi level changes accordingly to the surface dipole and leads to different charge transfer.

processes, such as the oxygen evolution reaction in solid oxide electrolysis cells, we believe that this concept possibly also applies to the effects of coatings on battery materials surfaces and could even be utilized to modify the surfaces of semiconductors for micro- and nanoelectronics.

## Methods

### Sample preparation and i-PLD

Prior to i-PLD investigations, Ti/Pt grids (5/100 nm) were prepared by photolithography and metal sputtering on both sides of (001) oriented yttria-stabilized zirconia single crystals (5 x 5 x 0.5 mm$^3$, Crystec GmbH). Subsequently, nanoporous La$_{0.6}$Sr$_{0.4}$CoO$_{3-\delta}$ electrodes were prepared on one side of the substrates by pulsed laser deposition with a KrF excimer laser (Compex Pro 201F, 248 nm, Coherent). Counter electrode depositions were performed at 450 °C substrate temperature, 0.4 mbar O$_2$ background pressure, 5.0 cm substrate-target distance, 1.1 J/cm$^2$ fluence at the target and a frequency of 5 Hz, producing kinetically fast electrodes with high internal porosity[37]. On the free current collecting grid, the investigated electrode materials were deposited during i-PLD[38–40]. During i-PLD, thin films and decorations were deposited at 600 °C, a background pressure of 0.04 mbar O$_2$, a substrate-target distance of 6.0 cm, 1.1 J/cm$^2$ laser fluence at the target and a frequency of 1 Hz. The samples were contacted with a Pt/Ir needle and the temperature was controlled via the ohmic offset in the impedance spectra that arises from the electrolyte conductivity whose temperature dependence is well investigated[41]. Targets for thin film deposition were prepared via a modified Pecchini route (LSC, LSF, PCO) or solid state synthesis (STF)[19]. Targets for decorations were pressed from the corresponding powders and sintered. For SrO, it was necessary to calcinate the powder in pure O$_2$ flow at 1200 °C for 12 h before pressing and a second 12 h sintering step at 1300 °C. It was also necessary to store the SrO target in a humidity-free environment (evacuated dessicator or glovebox to avoid hydroxide formation). Deposition rates during PLD were determined with a quartz balance inside the chamber. SnO$_2$ and SrO as decoration materials were chosen to span a broad range of acidity, while being manageable in terms of preparation (strong hydroxide and carbonate formation in air for more basic oxides, such as BaO) and contamination (volatile compounds for more acidic oxides, such as WO$_3$).

For work function investigations of pristine and decorated LSC thin films, substrates with an oxygen ion buffering counter electrode were fabricated as follows: 80 % wt. of 10 % Gd-doped ceria (GDC10, Treibacher) and 20 % wt. Fe$_2$O$_3$ (99.95 %, Merck) were dissolved in ink vehicle (Fuelcellmaterials, USA) in a 1:1 weight ratio and homogenized in a rotary ball mill (Fritsch pulverisette). The paste was spin-coated on the unpolished side of 5 × 5 × 0.5 mm$^3$ (001)-oriented single-crystalline YSZ substrates at a speed of 2400 RPM, and dried. After application of a thin layer of Pt paste (Tanaka), the counter electrode was sintered at 1050 °C for 3 hours in air. LSC thin films were prepared as described above.

### X-ray photoelectron spectroscopy

XPS measurements were conducted in two different setups. Near-ambient-pressure XPS to investigate the effect of acidic adsorbates was performed in a lab-based setup equipped with a PHOIBOS NAP photoelectron analyzer (SPECS, Germany) as well as a monochromated Al K-$\alpha$ XR 50 microfocus X-ray source. Cells were mounted in a custom-designed sample holder with a 4.5 x 4.5 mm$^2$ square hole for near-IR diode laser heating[42]. The sample was contacted with Pt-Ir wires for temperature control via in-situ measurement of the YSZ electrolyte conductivity. During NAP-XPS measurements, samples were exposed to 1 mbar O$_2$ and a temperature of 600 °C. Spectra were collected at an analyzer pass energy of 30 eV. For work function measurements, a sample bias of -20 V was applied to accelerate very low kinetic energy electrons towards the analyzer. Here, an analyzer pass energy of 5 eV

and an energy step width of 0.02 eV were used for better energy resolution. Due to the low electron kinetic energy and the long pathway to the analyzer, a precisely tuned active magnetic shielding was necessary.

In order to avoid work function changes through ubiquitous sulfur impurities in the atmosphere, we used a UHV-based XPS spectrometer to measure transition metal oxidation states and work function changes of pristine and decorated LSC samples. Cells with an oxygen ion buffering counter electrode (see above) were mounted in a PHI versaprobe 4 XPS spectrometer on a heated stage with electrical contacts for working and counter electrodes. The X-ray source was a monocromated Al-K-$\alpha$ source operating at 50 W, and photoelectrons were detected at an angle of 45° relative to the surface normal. In this geometry, 66 % of the signal originate from the topmost 1.6 nm, or 4 unit cells. The thin-film working electrode was kept at ground potential, while the electrochemical polarization was applied to the counter electrode by means of a Biologic SP-200 potentiostat. During XPS measurements, the temperature was controlled by evaluating the ionic conductivity of the YSZ single crystal[41].

For controlled oxygen activity in the working electrode, constant and known counter electrode oxygen activity is important. For this purpose, the sample was heated in the XPS chamber to 450 °C, and a constant current of +40 $\mu$A was applied to the cell for 2000 s, in order to release O$_2$ from the oxygen ion buffering counter electrode to the working electrode. During this time, the Fe$_2$O$_3$ phase of the counter electrode was partially reduced to a Fe/Fe$_{0.95}$O phase mixture. By this means, the counter electrode oxygen activity was fixed to that of the Fe/FeO phase equilibrium with

$$a(O_2)_{CE} = e^{\frac{2\Delta_f G^0}{RT}}, \quad (2)$$

where the standard formation enthalpy of FeO, $\Delta_f G^0$ is[43]

$$\Delta_f G^0 / J \cdot mol^{-1} = -303,097 + 683.5T - 91.4T \ln T + 0.05T^2. \quad (3)$$

Consequently, the oxygen activity in the LSC thin film working electrode can be determined from the cell voltage $U_{cell}$ according to Nernst's equation by

$$a(O_2)_{WE} = e^{\frac{2\Delta_f G^0 + 4FU_{cell}}{RT}}. \quad (4)$$

A similar cell design with an oxygen sub-stoichiometric Gd-doped ceria counter electrode was used already in a previous publication[44]. Here, the new counter electrode design gives an even better controllability of the oxygen activity. After the initial partial reduction of the counter electrode, 800 and 1000 mV were applied to a cell, resulting in oxygen activities of $1 \cdot 10^{-6}$ and $3.4 \cdot 10^{0}$ mbar, respectively.

Work functions were acquired by using the potentiostat in floating mode and applying -5 V to the working electrode vs. ground, followed by the acquisition of the secondary electron cutoff. XPS spectra were acquired at 27 eV analyzer pass energy and the secondary electron cutoff was measured at 13 eV pass energy.

### Computational methods

DFT-calculations were performed with the WIEN2k code[45] and the full-potential augmented plane wave plus local orbitals method[46,47]. All calculations were performed spin-polarized with the PBE-GGA (generalized gradient approximation) functional[48]. As both, LSC and PCO, exhibit correlated electron systems (3d electrons for Co and 4f electrons for Ce and Pr), we used a Hubbard U correction with potentials of 3.35 eV for Co[23,49], 5.0 eV for Ce and 4.5 eV for Pr[49,50]. Since PCO has not been the subject of many computational studies, the resulting density of states was compared with literature and was in very good agreement with previous reports, even though no spin-orbit coupling was used in this approach[50]. Volume optimizations were performed for LaCoO$_3$

and $CeO_2$, yielding lattice parameters of 3.83 and 5.44 Å, respectively, being in good agreement with experimental data[51,52]. As base model for LSC, we used a 2 x 2 x 5 (001) surface slab of $LaCoO_3$ and replaced all surface La and several bulk La atoms with Sr to simulate not only the LSC stoichiometry but also the SrO-termination which has been shown to be the equilibrium termination for LSC[53]. A symmetric slab was chosen to avoid overall dipole effects which might interact with surface decorations. While dipole correction schemes for asymmetric slabs are available[54,55], in this study, it is crucial to avoid undesired countercharge effects that might be introduced by nonideal correction. To minimize interactions between surface slabs, a 20 Å vacuum space was added. For PCO, we started with a 2 x 2 x 3 (111) surface slab of $CeO_2$ and replaced one surface Ce and one bulk Ce atom with Pr. The (111) orientation was chosen to avoid polar surfaces and because it is the lowest energy surface of ceria[56]. Full model cells with fractional coordinates are shown in supplementary note 9.

Building on these models, we added monolayers of the decoration materials. For LSC, SrO was added in a (001) oriented rock-salt structure, rotated by 45° relative to LSC, continuing the SrO termination. $SnO_2$ was added as a $BO_2$ plane of a perovskite structure, nominally leading to one unit cell of $SrSnO_3$ at the surface. For PCO, SrO was added as a (111) oriented monolayer with a reasonable match of the Sr-Sr bond length with the bulk value (3.86 Å vs. 3.65 Å experimental). As it proved energetically favorable, O atoms were gradually replaced by peroxide groups. $SnO_2$ was added continuing the (111) lattice of ceria. It is noteworthy that the atomic structure of the ultrathin decoration layer is expected to deviate significantly from the bulk material, and very idealized structures have been assumed for this study. For a true, quantitative computational treatment of surface dipole formation processes, atomically resolved and surface sensitive imaging methods are necessary to give conclusive evidence about the detailed structure of decoration layers.

To find energetically favorable configurations, structure relaxation was performed for all models, employing a Γ-centered 4 x 4 x 1 k-mesh with a -6.0 Ry energy separation between core and valence states. All final calculations were performed with the same basis-set size, determined by $R_{MT}^{min} K_{max} = 6$, with $R_{MT}$ being the smallest atomic sphere radius (2.20, 2.20, 1.87, 1.15, 1.38, 2.27, 2.27, 2.00 for La, Sr, Co, O, S, Ce, Pr and Sn, respectively) and $K_{max}$ being the largest reciprocal lattice vector. All relaxations were performed until residual forces did not exceed 3 mRy/bohr for three consecutive iterations. For charge redistribution investigations, Bader's quantum theory of molecules was used[20]. Here, a charge is assigned to every atom in the slab, whereas the atom is defined as the region bounded by a zero flux surface ($\vec{\nabla} \rho \cdot \vec{n} = 0$), and the electron density $\rho$ is integrated over this region. For investigations of adsorbed $O_2$ molecules, a surface oxygen atom was replaced with an $O_2$ molecule and a structure relaxation was performed. The coulomb potential was determined on a 100 x 100 x 1000 grid and averaged for each z-value.

## Data availability

All data used in this study as well as the structures that are discussed are available in the Materials Cloud database under accession code https://archive.materialscloud.org/deposit/records/2013.

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

## Acknowledgements

This research was funded in whole or in part by the Austrian Science Fund (FWF) grant 10.55776/P31654. For open access purposes, the author has applied a CC BY public copyright license to any author accepted manuscript version arising from this submission. The authors also acknowledge project funding by the Austrian Research Promotion Agency (FFG) for the project ELSA. M.S. gratefully acknowledges support from a Max Kade Fellowship of the Max Kade foundation. This research has received funding from the European Research Council (ERC) under the European Union's Horizon 2020 Research and Innovation programme, grant agreement no. 755744/ERC-Starting Grant TUCAS.

## Author contributions

M.S. and M.K. conceived the original idea for the project and planned the experiments. M.S. carried out all simulations and performed the electrochemical experiments. A.N. carried out all spectroscopic investigations. P.B. provided input for the simulations and J.F., M.K. and P.B. supervised the project. M.S. wrote the manuscript and A.N., C.R., P.B., J.F. and M.K contributed to the final version of the article.

## Competing interests

The authors declare no competing interests.

## Additional information

**Peer review information** : *Nature Communications* thanks Nikolai Tsvetkov, and the other, anonymous, reviewer(s) for their contribution to the peer review of this work. A peer review file is available.

