## [Peer Review File · Nature Communications]

REVIEWER COMMENTS

Reviewer #1 (Remarks to the Author):

The article, titled "Engineering Surface Dipoles on Mixed Conducting Oxides with Ultra-Thin Oxide Decoration Layers," explores the feasibility of altering the catalytic activity of oxide surfaces through the modification with ultra-thin layers. This modification leads to changes in surface dipoles and work function. Additionally, the authors propose considering the ionic potential of surface cations as a potential descriptor for these effects. Those are interesting and novel results. The paper presents both experimental and computational results, featuring high-quality discussions and can be considered for publication in Nature Communications after revisions.

1. The authors should include additional discussion regarding the rationale for considering ionic potential as a valuable descriptor. They might also explain why it is preferable over parameters such as surface acidity.
2. More comprehensive details on the electrochemical impedance spectroscopy (EIS) measurements are needed. It would be beneficial if the authors could incorporate EIS data, such as Nyquist plots. The paper mentions that the enhancement in catalytic activity results from improved charge transfer between the surface and absorbed oxygen. Do the EIS results support this conclusion?
3. The O1s spectra display a main peak at approximately 528 eV (see Fig. S2). However, this peak appears to lack perfect symmetry. It would be interesting to know if the authors observed any high-energy shoulder on the peak, similar to what has been reported in the literature (e.g., see DOI: 10.1002/adfm.202108005). A thorough analysis of the O peak could aid in identifying the concentration of defective oxygen or oxygen vacancy concentration (<https://doi.org/10.1002/er.4613>).

Reviewer #2 (Remarks to the Author):

This seems to be a highly interesting paper, but I realize that I do not have the expertise to evaluate the manuscript only after I have agreed on reviewing it. Specifically, I am not in the field of thin-film oxides for SOFC, nor do I perform DFT calculations. Basically, I am not familiar with the status quo, hence, am not in the position to evaluate the novelty of the work. However, I do have a couple of points for the authors to explain, perhaps, in their revision. (1) The concept of ionic potential, is averaged over all cationic sites on a surface, why is that so? does this mean that the anions on the surface are completely irrelevant as far as properties such as surface dipole and oxygen exchange are concerned? how would one treat host materials with different levels of anionic vacancies then? (2) Other than the XPS-based measurements of work function, is there any more characterizations? UPS for example to confirm the work function result, STM/STS for atomically resolved structure of adsorbates, or scanning Kelvin probe for combined imaging and work function measurements? Lack of structural characterization makes the conclusions a little unsettling.

Reviewer #3 (Remarks to the Author):

The authors present a comprehensive overview of the opportunities for engineering electroceramic materials through considerations of the Fermi energy. The manuscript establishes a solid foundation of the concept of the Fermi energy before exploring the routes of Fermi level engineering to design improved functional oxide materials. The manuscript is well written, and the figures are of excellent quality. I congratulate the authors on this excellent piece of work and recommend publication in JECR as submitted.

I noticed only two mistakes: 1) the references to Figure 5 in the text did not always match the mechanism being shown. 2) There is a missing 'about' in the first sentence of the final paragraph "Fermi level engineering is a way to think materials development differently."

Reviewer #4 (Remarks to the Author):

The paper is an extension of the acidity/basicity work previously presented by Nicollet et al. The unique ability of the authors to control the additive layer is interesting and the idea of interfacial dipole formation is intriguing. More information, however, and, in particular deeper contextualization/discussion of the findings is needed to make the manuscript publishable.

1) What is meant by strongly acidic and highly basic. This should be nuanced as SrO is less "basic" than e.g. Li₂O and SnO₂ is considered less "acidic" than e.g. WO₃. It should be more clearly justified why these two decorations were specifically selected?

2) Although the authors assert that the surface of the metal oxides has been decorated with a monolayer of either SnO₂ or SrO based on results previously attained on PCO, no morphological information is provided in the manuscript. It is unclear if there is verification for LSC. Ideally LEIS (the authors argue this method is suitable) for the samples shown in Fig.1 should be provided. Evidence is needed that a monolayer is formed and that no island/particle formation occurs.

3) The authors argue that SnO₂ is more acidic due to its high 4+ charge and small ionic radius while Sr is basic as it is only +2 and has a large radius. In addition to the more common SnO₂, SnO has been known to form, i.e. Sn would only have a +2 charge. As this is a key point, evidence is needed for the oxidation state of Sn, e.g. information possible from XPS? As a reference: work that discusses the transition of thin films from SnO to SnO₂ and or Sn the authors should refer to 10.1016/j.heliyon.2016.e00112.

Rationale for why XPS instead of the more common UPS was used to determine the work function.
<https://doi.org/10.1016/j.apsusc.2009.11.002>

4) Although covered in a previous paper, the authors should mention where the sulfur contamination is coming from during the XPS measurements. Is there a strong affinity for the entire LSC surface to S-species or do the authors believe that specifically e.g. the La-sites are affected?

5) What is meant by "chemically suitable configurations" were selected for the modelling?

6) Were the decorations added to both the top and the bottom of the slab for the calculation (images in supplementary information)? If so why?

7) The authors should elaborate what is meant by, "SrO-terminated LSC and SnO₂ terminated PCO show no significant tendency towards oxygen vacancy or peroxide formation - indicating that the proposed configuration likely reflects an energetically favorable structure." The authors also state that the formation of one surface vacancy is favorable for SnO₂ terminated LSC. Did the authors do energetic calculations for the adsorption of oxygen on the presented slabs? Were activation energies considered?

8) Calculating the density of states for transition metal oxides is not trivial. Furthermore, the description of heavier elements, such as the lower transition metals and lanthanides, typically involves relativistic effects, such as spin-orbit coupling. Specifically the small polaron conduction of Pr-doped Ceria is usually not correctly captured. The authors cite the work of Michel et al. 10.1021/acs.jpcc.0c05352 when discussing the use of Hubbard U correction. The DOS presented in Fig. 3f seems to have been done with the inclusion of spin-orbit coupling: This should be clarified!

9) The concepts of a surface dipole should be more clearly described. It could be useful to refer to the large body of work from PV on layered oxide structures for how best to approach such an explanation. A discussion similar in detail to that presented in the paper of Klein et al. (10.3390/ma3114892) would help the reader understand the suggested model. The authors specifically discuss how different sputtering treatments of SnO₂ films can influence surface dipoles. They also report that in some cases surface dipoles can be modified by post-deposition treatments (a point also briefly mentioned in the manuscript). This point goes back to the consideration of decorations that have multiple oxidation states (e.g. Sn). Do the authors have any insights into how robust the descriptor of ionic potential of surface cations is in systems with potentially varying oxidation states? This could be a particularly important consideration for the presented thin films as many of the surface metal sites are likely not stoichiometric. The authors may have made an effort to address this point in Figure 3, but the caption, " Next to SnO₂-decorated LSC (5.72 eV), the work functions of SnO_{1.75} (5.55 eV) and SnO (5.53 eV) decorations are shown additionally, as well as the work functions of SrO_{1.5} and SrO₂ decorations for SrO-decorated PCO." is difficult to understand!

10) From all the studied materials, LSF, STF, LSC and PCO, only LSC and PCO were studied in detail. A rationale for why these two samples were selected should be given.

The conclusion of the authors that it is unlikely that the decorations result in changed activation energy while asserting that it is likely that the energetics of the adsorbates are being influenced needs clarification. The authors argue that the surface potential will change the energy levels of adsorbed reaction intermediates, (e.g. peroxy or superoxol). Would this, however, not likely result in a change of the activation energy?

Minor point out of curiosity: The authors state that it is unlikely to observe molecular adsorbates directly during near ambient pressure XPS, due to their low concentration and because their signal would be masked by higher concentrated bulk oxygen and SO₂₋₄ adsorbates (unavoidable). Previously in the manuscript the authors argue that peroxy-species are likely present on the surface and visible in the XPS. What about the peroxy-species would likely make them visible in that particular case?

Response to the Reviewers Comments

Comments/Response/Changes

Reviewer #1 (Remarks to the Author):

The article, titled "Engineering Surface Dipoles on Mixed Conducting Oxides with Ultra-Thin Oxide Decoration Layers," explores the feasibility of altering the catalytic activity of oxide surfaces through the modification with ultra-thin layers. This modification leads to changes in surface dipoles and work function. Additionally, the authors propose considering the ionic potential of surface cations as a potential descriptor for these effects. Those are interesting and novel results. The paper presents both experimental and computational results, featuring high-quality discussions and can be considered for publication in Nature Communications after revisions.

We thank the Reviewer for their assessment of our manuscript. We have expanded the discussion according to the Reviewer's comments and have also added impedance spectroscopic investigations to the supplementary information (as we wanted to keep the manuscript concise, we chose this approach over adding a new chapter to the manuscript). We also expanded our XPS discussion. Please find our detailed response below.

1. The authors should include additional discussion regarding the rationale for considering ionic potential as a valuable descriptor. They might also explain why it is preferable over parameters such as surface acidity.

We introduced the ionic potential, as it is in our opinion a more fundamental quantity than the Smith Acidity, and, in contrast to the Smith acidity, is a property of the surface cation, and not of a binary oxide. We expect that a real decorated surface deviates from the bulk structure of the binary oxide and that the complex electrostatic situation at the surface of an oxide is indeed strongly dependent on the cations at the surface. In addition, the ionic radius is known for different oxidation states and offers higher flexibility compared to the Smith acidity, which is based on the determination of reaction enthalpies of selected binary oxides.

We expanded the discussion of surface descriptors in the S.I.:

In general, the comparison shows the expected result that the here explained metrics correlate very well with each other and are similarly well suitable to describe the properties of binary oxides. Differences emerge in their ease of use when considering more complicated oxides and in particular their surfaces. There, the Smith acidity, which is based on empirical data from binary oxides (and which also considers the structure of the products of reactions between binary oxides) is potentially not the best choice to describe the effects induced by modification processes. We suspect that real decorated surfaces deviate considerably from binary oxide structures and that it is the introduced cation that is decisive for the observed changes. Here we suggest that acidity and basicity (which are intuitive concepts and therefore desirable) are better correlated to other underlying ion-specific metrics such as the ionic

Response to the Reviewers Comments

potential or the electronegativity, which may be better suited to tackle more complicated problems. The ionic potential also has two further advantages: i) ionic radii are listed for different oxidation states, increasing the flexibility of the descriptor, ii) its constituents can be estimated by computational approaches, facilitating the synergy of experimental and theoretical studies. An overview of the quantitative correlations of the discussed metrics is given in the following figure (deviations for the ionic potential of SiO₂ might be due to the strong covalent character of the bond and our use of the formal 4+ charge):

2. More comprehensive details on the electrochemical impedance spectroscopy (EIS) measurements are needed. It would be beneficial if the authors could incorporate EIS data, such as Nyquist plots. The paper mentions that the enhancement in catalytic activity results from improved charge transfer between the surface and absorbed oxygen. Do the EIS results support this conclusion?

We added an additional section in the supporting information on decoration effects and their investigation with impedance spectroscopy. There, we summarize and refer to results published earlier that add more details on the electrochemical methods used. In particular, we added measurement results of impedance spectroscopy during the decoration of LSC64 and PCO10 with SrO and SnO₂, as these are the relevant material combinations presented in the main manuscript:

S.I.5. In-situ impedance spectroscopy during decoration

For a more in-depth discussion, we want to note here that detailed impedance spectroscopic investigations on thin films upon basic and acidic decoration have been presented by the authors in previous articles^{11,16,17}. Generally, all i-PLD measurements follow a similar procedure, which has been outlined in previous studies¹⁸. The investigated samples consist of a YSZ single crystal electrolyte, Ti/Pt current collecting grids on both sides of the substrate and a 200-300 nm nanoporous LSC64 counterelectrode on one side. On the other side, the working electrode is grown during i-PLD. The temperature during deposition is controlled very precisely via the ohmic offset, which contains the electrolyte resistance (that is well known from literature), as well as resistive contributions from wiring and the current collecting grids (that are measured beforehand). Resulting impedance spectra usually consist of three major contributions. i) the above mentioned ohmic offset, ii) a mid-frequency semicircle, which corresponds to the surface exchange resistance coupled with the chemical capacitance of the working electrode, and iii) a low-frequency arc, which corresponds to the surface exchange resistance coupled with the chemical capacitance of the counter electrode. The volume-related chemical capacitance of the counter electrode is much higher due to the higher thickness, leading to different characteristic frequencies of the two electrode contributions, and thus to well separable semicircles. It is worth mentioning, that for some samples, a small high-frequency shoulder appears, which is attributed to interfacial resistances between thin film and electrolyte.

As the working electrode feature is usually a nearly perfect semicircle, it is very unlikely that diffusion limitations affect the measurements (they would be visible as Warburg-type

Response to the Reviewers Comments

distortions on the high frequency side of the semicircle) and thus, the observed resistance is directly connected to the rate of the rate determining step of the oxygen exchange reaction. Our previous results have suggested that this rate determining step is related to charge transfer and dissociation¹⁹, but the details about this reaction mechanism are still not entirely clear. It is, however, very likely, that upstream reaction steps of the oxygen reduction reaction include adsorption of molecular oxygen and charge transfer onto the adsorbed oxygen.

During decoration, the only impedance contribution that is affected is the working electrode feature. Upon basic decoration, the surface exchange resistance decreases, suggesting faster oxygen exchange kinetics and vice versa for acidic decoration. In the figure below, exemplary impedance measurements of acidic and basic decoration for LSC64 and PCO are shown.

Fig. S7. a) Impedance spectra of $\text{La}_{0.6}\text{Sr}_{0.4}\text{CoO}_{3-\delta}$ in its pristine state as well as with 2 pls SrO and SnO_2 decoration. b) Impedance spectra of $\text{Pr}_{0.1}\text{Ce}_{0.9}\text{O}_{2-\delta}$ in its pristine state as well as with 2 pls SrO and SnO_2 decoration.

3. The O1s spectra display a main peak at approximately 528 eV (see Fig. S2). However, this peak appears to lack perfect symmetry. It would be interesting to know if the authors observed any high-energy shoulder on the peak, similar to what has been reported in the literature (e.g., see DOI: 10.1002/adfm.202108005). A thorough analysis of the O peak could aid in identifying the concentration of defective oxygen or oxygen vacancy concentration (<https://doi.org/10.1002/er.4613>).

We absolutely agree, that the O1s main peak might include information about oxygen lattice defects, and we considered this possibility in our interpretation. However, the situation here is slightly more complicated, as we deal with a perovskite with a metallic electronic structure. In fact, it is likely that this asymmetry is caused by the metallicity of LSC, as is also strongly indicated in the results of a previous study (10.1021/acs.jpcc.5b08596), where the asymmetry increases from $\text{SrTi}_{1-x}\text{Fe}_x\text{O}_{3-\delta}$ to $\text{La}_{1-x}\text{Sr}_x\text{FeO}_{3-\delta}$ to $\text{La}_{1-x}\text{Sr}_x\text{CoO}_{3-\delta}$. Hence, a lack of the asymmetry is observed for perovskites with a similar vacancy concentration but a lower conductivity and a less metallic electronic structure. In addition, the asymmetry does not change considerably when increasing the oxygen partial pressure during near-ambient-pressure XPS, as would be expected from the changing oxygen vacancy content.

We adapted the corresponding passage in the S.I.:

Response to the Reviewers Comments

The work function is highest on SnO₂ decorated LSC and lowest for SrO decorated LSC. The main oxygen 1s species exhibits some asymmetry which is attributed to the metal-like electronic structure of LSC (it has previously been shown that this peak asymmetry correlates with the metallicity of the electronic structure of perovskite oxides¹²).

Reviewer #2 (Remarks to the Author):

This seems to be an highly interesting paper, but I realize that I do not have the expertise to evaluate the manuscript only after I have agreed on reviewing it. Specifically, I am not in the field of thin-film oxides for SOFC, nor do I perform DFT calculations. Basically, I am not familiar with the status quo, hence, am not in the position to evaluate the novelty of the work. However, I do have a couple of points for the authors to explain, perhaps, in their revision.

(1) The concept of ionic potential, is averaged over all cationic sites on a surface, why is that so? does this mean that the anions on the surface are completely irrelevant as far as properties such as surface dipole and oxygen exchange are concerned? how would one treat host materials with different levels of anionic vacancies then?

(2) Other than the XPS-based measurements of work function, is there any more characterizations? UPS for example to confirm the work function result, STM/STS for atomically resolved structure of adsorbates, or scanning Kelvin probe for combined imaging and work function measurements? Lack of structural characterization makes the conclusions a little unsettling.

We thank the Reviewer for their careful evaluation of our manuscript, for their honesty regarding their review, and for their very interesting comments and remarks. We will discuss the two main concerns below.

- 1) The definition of the ionic potential we used in this study is the ratio of ionic charge and ionic radius. One convenient property of this descriptor is the fact that the ionic radii of cations are listed not only for different valence states but also for different coordination numbers. Thereby, different surface structures can be described with this metric. To account for more complex chemistries, we chose to average over the surface cationic sites as a first approximation. Hence, it allows to tentatively account for the differences between varying surface compositions, e.g. with a Sr-rich termination being more basic than a mixed surface with both, AO and BO₂ terminations. Regarding anion vacancies, on the one hand changing cation valences could account for the charge redistribution caused by these vacancies. On the other hand, we suspect that the surface vacancy concentration is actually related to the cation ionic potential, as it reflects the binding strength between oxygen and surface cations. We agree, however, that this metric is not able to quantify acidity or basicity changes due to stoichiometry changes on one and the same surface (e.g. caused by oxygen partial pressure changes). We rather see the metric as a means for a versatile comparison between different surface chemistries.

Response to the Reviewers Comments

We added the following passage to reflect this discussion in the manuscript (page 3):

It is worth mentioning, that averaging over the surface cation stoichiometry is only an estimate allowing for the comparison of different surface chemistries and does not account for the oxygen vacancy concentration at the surface as a function of temperature and oxygen partial pressure.

- 2) We agree with the Reviewer that we paid too little attention to a convincing structural characterization in the first version of the manuscript. At the same time, we ask the Reviewer to understand that it is not a trivial task to analyze the outermost surface chemistry of complex oxides at elevated temperatures. Before we detail the changes that we made to the manuscript, we will briefly discuss the different techniques mentioned by the Reviewer. It is true that UPS is a more common way to measure work functions, because the energy scale calibration with a Fermi edge kinetic energy of ~ 20 eV is more precise than that of an XPS spectrometer. However, the UPS spectrometer that is available to us is not compatible with sample heating and contacting options that are required for these experiments. Also, the photon source itself does not influence the sharpness or position of the secondary electron cutoff energy, as these secondary electrons always undergo many inelastic interactions. Consequently, work function differences found between the studied samples are equally accurate with XPS and UPS. However, the absolute values rely on a calibration standard (111-oriented Pt thin film). STM/STS and in particular the in-situ version of this technique (connected to the PLD chamber) would be the ideal tool to precisely characterize the studied surfaces and their evolution upon decoration. However, the characterization of thin film surfaces on the atomic scale with STM is best done on ideal surfaces such as the SrTiO₃ or La_{0.8}Sr_{0.2}MnO_{3- δ} surfaces that were studied in the group of Prof. Ulrike Diebold at TU Wien. The thin films studied in this work are by far not as ideal (as can also be seen in the AFM images that we added to the supporting information). Upon request at an earlier point during this study, Prof. Diebold predicted that it would take a substantial amount of time (at least several months) to reach a point of thin film quality, where she would expect publishable atomic scale STM results. We see this work as a proof of concept and we are convinced that in the future, atomic scale characterization combined with a fitting computational approach has strong potential to unveil even more details behind these complex but very powerful surface modifications.

The additional characterization that was accessible to us was atomic force microscopy and more low energy ion scattering (LEIS) measurements, and we added the corresponding new results of LSC and PCO surfaces to the supporting information. The AFM images show no differences between decorated and undecorated surfaces and hence, we can confidently exclude the possibility that the decoration agglomerates in the form of surface particles. While LEIS results of decorated PCO thin films suggest nearly full coverage with the decoration, for LSC, the host material cations are still visible on the surface, however, a stronger decoration ion peak appears,

Response to the Reviewers Comments

making for 56 % of the total surface cation signal (for a nominal decoration thickness of one monolayer). Based on these results, we conclude that the decoration grows as a very flat layer on the surfaces, eventually leading to a full coverage of the surfaces, but possibly not necessarily in layer-by-layer growth. We are aware that the models adopted in DFT calculations represent a very idealized system (also causing significantly stronger work function trends as observed experimentally) but considering the combination of all experimental and computational results, we are convinced that we successfully describe the fundamental properties of the investigated system.

The following section was added to the supporting information:

S. 7. Extended sample characterization

In Fig. S9, atomic force microscopy (AFM) images of LSC and PCO surfaces with different decorations are shown. These measurements were performed for previous publications¹¹ and showcase that the decorations do not lead to any particle formation on the surface, or in fact to any visible alteration of the surface at all. For LSC, polycrystalline thin films with SnO₂ and CaO decorations were investigated. In both cases, the granular surface of the LSC thin film is completely unchanged and no visible traces of the decoration (nominally one unit cell) can be seen on the surface. In the case of PCO, the thin films were deposited on a YSZ/GDC system, leading to epitaxial thin film growth, with atomic terraces being visible in AFM. Again, both SnO₂ and CaO decorations do not agglomerate, and the very flat surface remains unchanged during decoration (for SnO₂ decoration, LEIS measurements found a coverage of >90% of the surface cations with Sn, see below). Some few isolated particles are found on the surfaces, which are attributed to dirt on the sample surface.

Fig. S9. AFM images of LSC and PCO surfaces upon decoration with SnO₂ and CaO. In both cases, no visible traces of the decoration can be seen in AFM images.

Response to the Reviewers Comments

In addition to AFM measurements, low energy ion scattering (LEIS) measurements yield further insight into the chemistry of the outermost surface of PCO and LSC thin films. In a previous publication¹¹, LEIS was used to investigate PCO decorated with SnO₂ and SrO. For one nominal unit cell of SrO, no Ce or Pr signal could be identified during LEIS measurements, for SnO₂, the Sn signal amounts to around 90 % of the signal, indicating a high degree of coverage. For LSC, a SnO₂ (1 ML) decorated sample was investigated with a 5 keV ²⁰Ne⁺ primary analysis beam to allow for a proper separation of different cation signals. Again, the Sn signal is the largest signal and dominates the surface cation chemistry (56 % of the total cation signal), however, the coverage does not seem to be as complete as for the case of PCO. Considering the low amounts of deposited material and the unchanged morphology during AFM measurements, the experimental evidence conclusively indicates that the decorations grow very flat and ultimately lead to full coverage, but they do not necessarily grow in a layer-by-layer growth mode. In the case of PCO, also depth profiles were recorded with LEIS¹¹, showing a sharp decrease of the decoration ion signal below the surface, showing no significant interdiffusion into the host material.

To combine chemical analysis with lateral resolution, we also performed a secondary ion mass spectrometry measurement on a 100 x 100 μm² area of SnO₂ decorated LSC. The measurement was performed in the collimated burst alignment (CBA) mode (10.1039/C3JA50059D) which allows for high lateral resolution. Again, an overlay of the Sn and the Sr signal (probing depth around 1 nm) reveals no noticeable agglomeration in larger particles or islands.

Response to the Reviewers Comments

The combination of these results emphasizes the complexity of these surfaces, particularly at high temperatures, but also showcases that the decoration layers prefer a flat growth mode vs. island or particle growth. This also is reflected in the XPS results, which probe the average over the whole surface. The fact that XPS trends are not as pronounced as the corresponding DFT model calculations (in particular for LSC) might indicate that we indeed do not deal with a perfectly homogeneous monolayer coverage but with a somewhat nonideal coverage, i.e. only the majority of the surface is covered small amounts of the host material remain exposed at the surface.

Reviewer #3 (Remarks to the Author):

The authors present an excellent study on the effect of ultra-thin oxide layers on the surface of mixed-conductors. Interestingly they proposed a new descriptor, the ionic potential, to describe how the work function and the surface exchange kinetics can be altered depending on the decoration used. I think this work provided valuable insights into surface behaviour and will be of great interest to those working in the field of mixed-conductors as well as the broader scientific community. The experiments and simulations appear to have been meticulously carried out, the manuscript is well-written, and the data is well-presented. I've carefully read through the manuscript a couple of times, and I did not find anything that I believe needs to be changed. As such, I recommend this work to be published as-is. I would like to congratulate the authors on an excellent and very interesting study.

We thank the Reviewer for their assessment of our manuscript and are delighted that they find the results interesting and insightful.

Reviewer #4 (Remarks to the Author):

The paper is an extension of the acidity basicity work previously presented by Nicollet et al. The unique ability of the authors to control the additive layer is interesting and the idea of interfacial dipole formation is intriguing. More information, however, and, in particular deeper contextualization/discussion of the findings is needed to make the manuscript publishable.

Response to the Reviewers Comments

We thank the Reviewer for their careful evaluation of our manuscript and will discuss the comments point by point below. We are grateful for this review, as it constructively challenged our work with very valid arguments and gave us the opportunity for a thorough improvement of our manuscript with additional data and revised discussions, which ultimately led to a better explanation of our experimental and computational results.

- 1) What is meant by strongly acidic and highly basic. This should be nuanced as SrO is less “basic” than e.g. Li_2O and SnO_2 is considered less “acidic” than e.g. WO_3 . It should be more clearly justified why these two decorations were specifically selected?

This assignment was based on the Smith acidity scale, and we agree that it is beneficial to put the terms “strongly basic” and “strongly acidic” into perspective. We adjusted the text correspondingly. These oxides were chosen, as they span the broadest acidity range while still being relatively easy to work with and to deposit with PLD. More acidic oxides, such as MoO_3 or WO_3 , might contaminate the PLD chamber and using more basic oxides, such as BaO was attempted, but these were unfortunately not stable during target synthesis. We added this justification to the manuscript.

We provided additional information about the oxides in the introduction (page 1):

We probed the effects of SrO, a highly basic oxide (-9.4 on the Smith acidity scale, similar basicity as Li_2O at -9.2), and SnO_2 , a strongly acidic oxide (2.2 on the Smith acidity scale, being slightly less acidic than WO_3 at 4.7), on the work function and the electronic structure of $\text{La}_{0.6}\text{Sr}_{0.4}\text{CoO}_{3-\delta}$ (LSC) and $\text{Pr}_{0.1}\text{Ce}_{0.9}\text{O}_{2-\delta}$ (PCO), two chemically and structurally different MIEC oxides.

We expanded the sample preparation section (page 7):

SnO_2 and SrO as decoration materials were chosen to span a broad range of acidity, while being manageable in terms of preparation (strong hydroxide and carbonate formation in air for more basic oxides, such as BaO) and contamination (volatile compounds for more acidic oxides, such as WO_3).

- 2) Although the authors assert that the surface of the metal oxides has been decorated with a monolayer of either SnO_2 or SrO based on results previously attained on PCO, no morphological information is provided in the manuscript. It is unclear if there is verification for LSC. Ideally LEIS (the authors argue this method is suitable) for the samples shown in Fig.1 should be provided. Evidence is needed that a monolayer is formed and that no island/particle formation occurs.

We fully agree with the Reviewer that we did not discuss the sample morphology satisfyingly in the original manuscript. We didn't expect it to be advisable to perform LEIS on the same samples that were measured in XPS, as several temperature and voltage treatments were performed on these samples, and these treatments will likely have altered the surface chemistry considerably. We therefore performed further LEIS measurements on LSC

Response to the Reviewers Comments

decorated with SnO₂, which we added to a new section (S.7. Extended sample characterization) in the supporting information, together with atomic force microscopy images of decorated LSC and PCO. The LEIS spectra on LSC still show some signals of the host materials cations, which are overshadowed by a strong Sn signal. The same sample was also measured by AFM to investigate possible particle or island formation. Since AFM results clearly show that the morphology is completely unaltered, we can exclude the agglomeration to particles or distinct islands on the surface. We see with LEIS, however, that the coverage is not 100%, and it seems that it is less than for the same decoration amount on PCO. Considering that the Sn signal still amounts to 56% of the total cation signal in LEIS measurements, we conclude that the decoration layer forms in a very flat manner, ultimately leading to a full coverage (as seen for PCO) but not necessarily in a layer-by-layer growth. We believe that this is also reflected by our XPS results, which show weaker trends than are predicted by DFT calculations. As XPS takes an area-weighted average over local work functions for a larger spot size (10.1103/PhysRevApplied.19.037001), the possibility of a not fully covering decoration layer must be considered in the interpretation of our results. For further experimental evidence, we also performed secondary ion mass spectrometry measurements on a SnO₂ decorated LSC thin film in collimated burst alignment (CBA) mode, which allows for relatively high spatial resolution. On a 100 x 100 μm² scan, no signs of island formation or spatial inhomogeneity could be detected. We concretized the manuscript in critical passages and paid more attention to an accurate description of our surfaces.

We adapted the text on page 2:

As low energy ion scattering (LEIS) measurements and atomic force microscopy images show very high surface coverage for PCO and LSC thin films decorated with SrO and SnO₂¹⁴ as well as no signs of island or particle growth (see S.7 in the supporting information), we assume a flat growth mode of the decoration layer which ultimately leads to a homogenous coverage. Computational results as well as i-PLD experiments support this assumption (see below). While XPS measurements probe a certain sample volume and cannot distinguish between island growth and full layer coverage, LEIS truly probes the outermost surface chemistry and is thus an invaluable tool to investigate modified surfaces.

We added a discussion about the link between computational and experimental results on page 5:

At this point, we also want to link computational results to the real surface morphology. Based on our surface analysis with LEIS and AFM the decoration layers grow very flat and do not form islands or particles, but possibly also do not cover the surface fully for thin decoration layers (see S.7. in the supporting information). Comparing experimental work function measurements with DFT predictions, we observed that the experimentally found trend is indeed weaker than predicted by DFT (especially for LSC, which seems to exhibit a higher fraction of free surface). As XPS work function measurements take an area-weighted average over the work function in a certain spot size²³ and thus might also record signal from small amounts of undecorated surface, this is in good agreement with the surface morphologies and chemistries of our samples.

Response to the Reviewers Comments

The following section was added to the supporting information:

S. 7. Extended sample characterization

In Fig. S9, atomic force microscopy (AFM) images of LSC and PCO surfaces with different decorations are shown. These measurements were performed for previous publications¹¹ and showcase that the decorations do not lead to any particle formation on the surface, or in fact to any visible alteration of the surface at all. For LSC, polycrystalline thin films with SnO₂ and CaO decorations were investigated. In both cases, the granular surface of the LSC thin film is completely unchanged and no visible traces of the decoration (nominally one unit cell) can be seen on the surface. In the case of PCO, the thin films were deposited on a YSZ/GDC system, leading to epitaxial thin film growth, with atomic terraces being visible in AFM. Again, both SnO₂ and CaO decorations do not agglomerate, and the very flat surface remains unchanged during decoration (for SnO₂ decoration, LEIS measurements found a coverage of >90% of the surface cations with Sn, see below). Some few isolated particles are found on the surfaces, which are attributed to dirt on the sample surface.

Fig. S9. AFM images of LSC and PCO surfaces upon decoration with SnO₂ and CaO. In both cases, no visible traces of the decoration can be seen in AFM images.

In addition to AFM measurements, low energy ion scattering (LEIS) measurements yield further insight into the chemistry of the outermost surface of PCO and LSC thin films. In a previous publication¹¹, LEIS was used to investigate PCO decorated with SnO₂ and SrO. For one nominal unit cell of SrO, no Ce or Pr signal could be identified during LEIS measurements, for SnO₂, the Sn signal amounts to around 90 % of the signal, indicating a high degree of coverage. For LSC, a SnO₂ (1 ML) decorated sample was investigated with a 5 keV ²⁰Ne⁺ primary analysis beam to allow for a proper separation of different cation signals. Again, the Sn signal is the largest signal

Response to the Reviewers Comments

contribution and dominates the surface cation chemistry (56 % of the total cation signal), however, the coverage does not seem to be as complete as for the case of PCO. Considering the low amounts of deposited material and the unchanged morphology during AFM measurements, the experimental evidence conclusively indicates that the decorations grow very flat and ultimately lead to full coverage, but they do not necessarily grow in a layer-by-layer growth mode. In the case of PCO, also depth profiles were recorded with LEIS¹¹, showing a sharp decrease of the decoration ion signal below the surface, showing no significant interdiffusion into the host material.

To combine chemical analysis with lateral resolution, we also performed a secondary ion mass spectrometry measurement on a 100 x 100 μm^2 area of SnO₂ decorated LSC. The measurement was performed in the collimated burst alignment (CBA) mode (10.1039/C3JA50059D) which allows for high lateral resolution. Again, an overlay of the Sn and the Sr signal (probing depth around 1 nm) reveals no noticeable agglomeration in larger particles or islands.

The combination of these results emphasizes the complexity of these surfaces, particularly at high temperatures, but also showcases that the decoration layers prefer a flat growth mode vs. island or particle growth. This also is reflected in the XPS results, which probe the average over the whole surface. The fact that XPS trends are not as pronounced as the corresponding DFT model calculations (in particular for LSC) might indicate that we indeed do not deal with a perfectly homogeneous

Response to the Reviewers Comments

monolayer coverage but with a somewhat nonideal coverage, i.e. only the majority of the surface is covered small amounts of the host material remain exposed at the surface.

- 3) The authors argue that SnO₂ is more acidic due to its high 4+ charge and small ionic radius while Sr is basic as it is only +2 and has a large radius. In addition to the more common SnO₂, SnO has been known to form, i.e. Sn would only have a +2 charge. As this is a key point, evidence is needed for the oxidation state of Sn, e.g. information possible from XPS? As a reference: work that discusses the transition of thin films from SnO to SnO₂ and on Sn the authors should refer to [10.1016/j.heliyon.2016.e00112](https://doi.org/10.1016/j.heliyon.2016.e00112). Rationale for why XPS instead of the more common UPS was used to determine the work function. <https://doi.org/10.1016/j.apsusc.2009.11.002>

Considering that the kinetic effect of SnO_x decoration on the oxygen exchange reaction and on the work function is in good agreement with its acidity according to the Smith acidity scale, we deem it very unlikely that it is actually present as SnO. In addition, Sn⁴⁺ is the most stable oxidation state for Sn and preferred in the predominant number of compounds. We can of course not exclude the possibility of mixed oxidation states of Sn on the surface, and consequentially also the possibility of deviation from the perfect SnO₂ stoichiometry. We revisited our XPS measurements, and a single Sn 3d 5/2 peak appears at 485.8 eV, which is at the lower end for SnO₂ and the upper end for SnO. Ultimately, computational and experimental results point towards a rather acidic SnO_x decoration, but again, we want to emphasize, that for a quantitative and thorough understanding of the electronic effects induced by such ultrathin decorations, precise atomic-scale surface characterization and corresponding computational models are necessary.

We extended our discussion on page 4:

At this point, we want to emphasize that, similarly to our computational models, also the actual stoichiometry of the deposited decoration layer is unclear (and hardly accessible for analytic tools due to the tiny sample volume). While SnO₂ decoration leads to a strong decrease of oxygen exchange kinetics, as would be predicted by its Smith acidity, we cannot exclude the possibility of mixed oxidation states or deviation from ideal oxygen stoichiometry (i.e. SnO_{2-δ}).

Regarding the choice of XPS vs. UPS measurements, we mainly chose XPS because the UPS spectrometer that is available to us is not compatible with sample heating and contacting options that are required for these experiments. Also, the photon source itself does not influence the sharpness or position of the secondary electron cutoff energy, as these secondary electrons always undergo many inelastic interactions. Consequently, work function differences found between the studied samples are equally accurate with XPS and UPS. However, the absolute values rely on a calibration standard (111-oriented Pt thin film).

Response to the Reviewers Comments

- 4) Although covered in a previous paper, the authors should mention where the sulfur contamination is coming from during the XPS measurements. Is there a strong affinity for the entire LSC surface to S-species or do the authors believe that specifically e.g. the La-sites are affected?

Generally, we observe that LSC is much more susceptible towards sulphate formation than e.g. PCO, which aligns also with LSC surfaces being more basic than PCO surfaces. Since LSC has been experimentally shown to be largely SrO terminated, we believe that sulphate groups predominantly form with Sr cations. This is also supported by Sr- and S-rich particles that are found on top of the LSC surface after long-term exposure to lab air. Regarding the origin of S during XPS measurements, we added a corresponding text passage at the respective location on page 3:

Upon exposure of pristine surfaces to considerable gas phase pressures ($>10^{-3}$ mbar), a new oxygen species appears at 532 eV, alongside a weak corresponding S species, **that is caused by the inherent impurity content of measurement atmospheres despite using nominally high purity gases²³.**

- 5) What is meant by “chemically suitable configurations” were selected for the modelling?

This phrasing was indeed unfortunate. We adjusted the respective text passage on page 3:

As a proof of concept, we selected geometric configurations for SrO and SnO₂ layers on LSC and PCO based on appropriate bond distances and orientation with regard to the host lattice as a starting point for structural relaxation (for more details refer to S.8. in the supporting information).

- 6) Were the decorations were added to both the top and the bottom of the slab for the calculation (images in supplementary information)? If so why?

Yes, the decorations were added to both the top and the bottom of the slab for the calculations to avoid an overall dipole in the supercell and eventual inaccuracies in the calculation of the work function. This is explained in the Methods section on page 8:

“A symmetric slab was chosen to avoid overall dipole effects which might interact with surface decorations.”

- 7) The authors should elaborate what is meant by, “SrO-terminated LSC and SnO₂ terminated PCO show no significant tendency towards oxygen vacancy or peroxide formation - indicating that the proposed configuration likely reflects an energetically favorable structure.” The authors also state that the formation of one surface vacancy is favorable for SnO₂ terminated LSC. Did the authors do energetic calculations for the adsorption of oxygen on the presented slabs? Were activation energies considered?

Response to the Reviewers Comments

We added a more precise explanation to the corresponding text passage on page 3:

SrO-terminated LSC and SnO₂ terminated PCO show no significant tendency towards oxygen vacancy or peroxide formation, i.e. removing a neutral oxygen atom or adding a second oxygen atom to a surface oxygen (and thus forming a peroxide) does not lead to total energy gains - indicating that the proposed configuration likely reflects an energetically favorable structure.

Regarding energetic calculations for adsorbed oxygen, we did not attempt any barrier calculations or reaction step simulations in this study. In a previous study, we did see substantial changes of idealized barriers for the adsorption of molecular O₂ into a surface vacancy upon SO₃ adsorbate formation (although the absolute values are not reliable, an increase of the barrier is very likely), so we would not be surprised to see a change in adsorption barriers on decorated surfaces as well due to surface charge formation. However, since the exact adsorption process on these decorated surfaces is unclear, we doubt the overall reliability of such calculations. We revisited the calculations we performed for the densities of state of molecular oxygen on PCO and compared the total energies to the ideal slab and molecular O₂. In agreement with our interpretation, the total energy decreases by 0.08 eV (more favorable) for O₂ on SrO-decorated PCO and increases by 0.01 eV (more unfavorable) for SnO₂ decorated PCO, confirming that basic decorations favor the presence of O₂ adsorbates. However, we believe that for a true energy diagram, a thorough mechanistic analysis with the calculation of different reaction pathways and scenarios would be necessary, which goes far beyond the scope of this study. In addition, reliable energy evaluations would require knowledge of the energetically favorable stoichiometry, which again would require much more detailed insight into the atomic scale structure of the surface decorations.

8) Calculating the density of states for transition metal oxides is not trivial. Furthermore, the description of heavier elements, such as the lower transition metals and lanthanides, typically involves relativistic effects, such as spin-orbit coupling. Specifically the small polaron conduction of Pr-doped Ceria is usually not correctly captured. The authors cite the work of Michel et al. 10.1021/acs.jpcc.0c05352 when discussing the use of Hubbard U correction. The DOS presented in Fig. 3f seems to have been done with the inclusion of spin-orbit coupling: This should be clarified!

We fully agree with the Reviewer, and we performed the calculations carefully and in consideration of previous literature results. The choice of the U values relied on literature in this case and in the case of PCO, our calculations were indeed guided by the paper the Reviewer is referring to. Our DOS of PCO, which was evaluated from a spin-polarized calculation with GGA+U without spin-orbit coupling is in very good agreement with the ones the authors present in the mentioned paper (10.1021/acs.jpcc.0c05352) with PCO showing semiconducting behavior with Pr4f intragap states and an O2p-dominated valence band. In addition, as we do not introduce additional electrons (e.g. by O bulk nonstoichiometry), we believe that our results properly reflect the electronic structure of PCO.

Response to the Reviewers Comments

We added the following text passage to the methods section:

Since PCO has not been the subject of many computational studies, the resulting density of states was compared with literature and is in very good agreement with previous reports, even though no spin-orbit coupling was used in this approach⁴⁴.

- 8) The concepts of a surface dipole should be more clearly described. It could be useful to refer to the large body of work from PV on layered oxide structures for how best to approach such an explanation. A discussion similar in detail to that presented in the paper of Klein et al. (10.3390/ma3114892) would help the reader understand the suggested model. The authors specifically discuss how different sputtering treatments of SnO₂ films can influence surface dipoles. They also report that in some cases surface dipoles can be modified by post-deposition treatments (a point also briefly mentioned in the manuscript). This point goes back to the consideration of decorations that have multiple oxidation states (e.g. Sn). Do the authors have any insights into how robust the descriptor of ionic potential of surface cations is in systems with potentially varying oxidation states? This could be a particularly important consideration for the presented thin films as many of the surface metal sites are likely not stoichiometric. The authors may have made an effort to address this point in Figure 3, but the caption, "Next to SnO₂-decorated LSC (5.72 eV), the work functions of SnO_{1.75} (5.55 eV) and SnO (5.53 eV) decorations are shown additionally, as well as the work functions of SrO_{1.5} and SrO₂ decorations for SrO-decorated PCO." is difficult to understand!

We thank the Reviewer for this comment and have expanded our discussion of the surface dipole. Indeed, the concept of surface dipoles has been widely discussed in different contexts, in particular with regard to surface reduction/oxidation, as suggested by the Reviewer, and also with regard to the formation of surface adsorbates in contact with a given atmosphere. We attempted a more comprehensive description of the surface dipole, which, in this case, is even more complicated as it also entails the heterointerface between host material and decoration. We want to emphasize, that this study doesn't aim for a structurally exact representation of the systems, as this would require atomic-scale knowledge of these surfaces, which requires a much more sophisticated surface analysis as is available to us at this moment. However, we are confident, that this study reveals the fundamental qualitative interactions underlying surface decorations on mixed conducting oxides and will be the basis for future studies unveiling the exact details of these processes.

We expanded the discussion on page 5:

Both mechanisms contribute to the buildup of electric fields on the surface, consequently leading to changes in the work function. At this point, it is important to emphasize, that this analysis was done for idealized, stoichiometric monolayers, showcasing fundamental principles of surface dipole formation. As we deal with mixed ionic and electronic conducting materials at high temperatures and in contact with the gas phase, real interfaces likely exhibit higher complexity.

Response to the Reviewers Comments

From an electrochemical point of view, dipole emergence is caused by different electrochemical potentials of electrons in the decoration and the material. Specifically, in the exemplary case of a negative surface charge, the electrochemical potential of the electron initially lies higher than for the decoration. For mixed conductors, depending on a balance between changed defect formation energies (which are also encoded in the electronic structure, e.g. via the O-2p band center²¹) and electrostatic potential buildup, complicated space charge zones might form at high temperatures, entailing altered oxygen vacancy concentrations, nonstoichiometric decoration layers, and complicated reconstructions. Also intermixing between decoration and host material cannot be excluded after longer periods at high temperature. Partly, this is already indicated by our analysis of varying oxygen content in decorations shown in Fig. 3 a), where changing oxygen stoichiometry also leads to changes in the work function (examples of the interplay of stoichiometry and surface dipole have also been previously discussed in literature²²). In addition, the formation of adsorbates on a decoration in contact with a gas phase will also affect the surface dipole as well as the space charge layer and induce further charge transfer. An extended discussion of the electrochemical potential landscape for the example of SrO decoration on LSC is given in S.4. of the supporting information.

We also adjusted the mentioned caption:

a) Ab-initio work functions of LSC and PCO surfaces, pure or modified with SrO, SnO₂ and SO₃. In the cluster of values for SnO₂-decorated LSC (5.72 eV), also the work functions of SnO_{1.75} (5.55 eV) and SnO (5.53 eV) decorations are shown additionally.

- 9) From all the studied materials, LSF, STF, LSC and PCO, only LSC and PCO were studied in detail. A rationale for why these two samples were selected should be given. The conclusion of the authors that it is unlikely that the decorations result in changed activation energy while asserting that it is likely that the energetics of the adsorbates are being influenced needs clarification. The authors argue that the surface potential will change the energy levels of adsorbed reaction intermediates, (e.g. peroxo or superoxol). Would this, however, not likely result in a change of the activation energy?

LSC and PCO were studied because they represent two fundamentally different material classes, perovskites and fluorites, and share no common cations. Both exhibit fast oxygen exchange kinetics (in particular in the case of LSC) and both accommodate substantial oxygen vacancy concentrations. Furthermore, i-PLD experiments have shown that various perovskite materials exhibit similar oxygen partial pressure and temperature dependences of their oxygen exchange kinetics, emphasizing the overall similarity of their pristine surfaces. We provided more details about our reasoning in the revised version:

We probed the effects of SrO, a highly basic oxide (-9.4 on the Smith acidity scale, similar basicity as Li₂O at -9.2), and SnO₂, a strongly acidic oxide (2.2 on the Smith acidity scale, being slightly less acidic than WO₃ at 4.7), on the work function and the electronic structure

Response to the Reviewers Comments

of $\text{La}_{0.6}\text{Sr}_{0.4}\text{CoO}_{3-\delta}$ (LSC) and $\text{Pr}_{0.1}\text{Ce}_{0.9}\text{O}_{2-\delta}$ (PCO), mixed conducting representatives of two fundamentally different material classes, perovskite and fluorite oxides.

Regarding the second part of this question, the answer is more complicated. Analyzing the available literature on surface infiltration and decoration based on the acidity/basicity concept (which has mostly been done by Nicollet et al., Seo et al. and the authors), the studies seem to agree that the activation energy remains largely unaffected by the surface modification. It is in fact, an unresolved question, how a decoration can change the oxygen exchange kinetics without significantly altering its activation energy. One possibility in our opinion is the proposed idea that the decoration lowers the energy level of specific reaction intermediates. If for example, the rate determining step of the reaction would be the dissociation of charged molecular oxygen adsorbates, and a basic decoration would favor the presence of these adsorbates energetically, it would be possible that the kinetic barrier for the rate determining step remains approximately the same. This idea seems also to be supported by our computational analysis. Furthermore, we see this theory substantiated by the fact that experimental evidence for peroxo-species (as described in the next comment/answer) could only be found for LSC with a basic surface decoration. We want to stress, that the discussion about the activation energy and the kinetic impact of surface decorations on the oxygen exchange reaction is still unresolved and we believe that the experimental results we present in this study will help to build a better understanding of this problem.

Minor point out of curiosity: The authors state that it is unlikely to observe molecular adsorbates directly during near ambient pressure XPS, due to their low concentration and because their signal would be masked by higher concentrated bulk oxygen and SO_2^{-4} adsorbates (unavoidable). Previously in the manuscript the authors argue that peroxo-species are likely present on the surface and visible in the XPS. What about the peroxo-species would likely make them visible in that particular case?

We thank the Reviewer for this remark, we did not communicate our findings clearly enough. During standard near ambient pressure XPS, peroxo-species will not be visible due to two circumstances. On the one hand, oxygen partial pressures during NAP-XPS are limited to the mbar range. On the other hand, unavoidable SO_4 adsorbates cause an oxygen signal at the same energy where we expect peroxo-species. During this study, we performed the first attempts of a new measurement approach during XPS. We performed experiments on SrO decorated LSC in UHV and at relatively low temperature (450 °C), such that oxygen exchange is strongly inhibited. By applying a controlled amount of bias voltage vs. an Fe/FeO oxygen reservoir on the counter electrode side, we were able to increase the effective oxygen partial pressure in the working electrode beyond the mbar range and to flood the surface with oxygen. Thereby, we could substantially increase the concentration of the desired peroxo-species at the LSC surface without the presence of acidic impurity species and investigate the effect on the oxygen XPS signal. Here, we suspect that peroxo-species cause an O1s species at around 532 eV, whose intensity can be tuned by adjusting the bias voltage and thus the species concentration on the surface.

We added a more detailed explanation to chapter S.I.2:

Response to the Reviewers Comments

Critical to this approach is the fact that measurement are performed in UHV and also at relatively low temperatures (450 °C) Thereby, the surface exchange is very slow and anodic polarization leads to a very high oxygen chemical potential at the working electrode surface, facilitating the formation of these peroxide adsorbates. In addition, the XPS signature is visible in UHV, because no SO_4^{2-} adsorbates are present on the surface in these conditions.

REVIEWER COMMENTS

Reviewer #1 (Remarks to the Author):

After revision paper is suitable for publication.

Reviewer #2 (Remarks to the Author):

The authors have addressed the concerns I raised in the previous review. The paper can be published as it is now.

Reviewer #4 (Remarks to the Author):

Overall the manuscript is greatly improved! Two points still remain in which further clarification would be useful.

1. Relates to original comment 8: The DOS and band diagrams shown in Fig. 4 are useful as they provide a simplified depiction of the situation from which discussion can start. It is clear that these are only model systems and that is it very challenging to capture the real situation. That being said Figure 4 provides a great deal of information in a highly condensed manner making it difficult for the reader to follow. It would be useful to do a more stepwise depiction and maybe to reconsider or at least critically discuss the selected situations.

The authors show the DOS of pristine PCO in Figure 3. They then show the DOS with the additives and considering adsorbed oxygen in Figure 4. It would be useful to show just the DOS with the additive as an intermediary step.

The authors have not introduced any additional electrons to the pristine PCO, as a result it is represented as an intrinsic semiconductor. In the depicted schematics, there are no electrons in the conduction band, the acceptor level (adsorbed oxygen) would then have to take electrons from the valence band. As a result, using this model, it would be expected that upon oxygen adsorption PCO, at least near the surface, shows p-type conduction. SrO is expected to insert electrons into the PCO, e.g. would insert electrons into the conduction band (must occur this way as no other free band is available). Alternatively, assuming this model, SnO₂ should extract electrons from the valence band again causing p-type surface near behaviour.

At the examined temperatures and oxygen conditions, PCO is known to be redox active. This has even been verified by the authors themselves

(<https://pubs.rsc.org/en/content/articlelanding/2022/TA/D1TA07128A>) The formation of oxygen vacancies are compensated by the reduction of Pr(IV) to Pr(III). The relatively low mobility of the electronic charge carriers is typically interpreted to indicate small polaron conduction, e.g. electrons in localized Pr4f bands. See paper by JJ Kim Chem. Mater. 2014, 26, 3, 1374–1379 In this case an acceptor would therefore likely remove electrons from the Pr4f band, while a donor would add electrons to it. This situation is more inline with the change in conduction described by Nicollet et al. previously,

decrease in conduction as a result of the acidic additive and increased conduction due to the basic additive. If possible, in consideration of this behaviour, could the situation be more accurately modelled? Maybe by using the DOS of Michel et al. with an additional electron (polaron) or by using a slab with defects as the starting point? The Ce4f band should also be considered, as an additional localized level energetically under the conduction band.

Minor point: As in order to be removed from the material the electron must cross the surface the vacuum level should not bend but simply shift.

2. Relates to original comment 9: It is still not clear how a lowering of the product energy (dissociated oxygen) would result in an increased rate, assuming that the activation energy and the starting state are the same. This seems like it would change the thermodynamics but not necessarily the kinetics.

3. Minor comment about former 6: The authors should add a bit more information about why the slab is symmetric making their reasoning stronger by citing relevant literature (e.g. Dipole correction for surface supercell calculations, Lennart Bengtsson, Phys. Rev. B 59, 12301 – Published 15 May 1999)

Response to the Reviewers Comments II

Comments/Response/Changes

Reviewer #4 (Remarks to the Author):

Overall the manuscript is greatly improved! Two points still remain in which further clarification would be useful.

We'd like to thank the Reviewer again for their careful evaluation of our revised manuscript. Please find our response to your comments in detail below, we hope that we can resolve all concerns of the Reviewer.

1. Relates to original comment 8: The DOS and band diagrams shown in Fig. 4 are useful as they provide a simplified depiction of the situation from which discussion can start. It is clear that these are only model systems and that is it very challenging to capture the real situation. That being said Figure 4 provides a great deal of information in a highly condensed manner making it difficult for the reader to follow. It would be useful to do a more stepwise depiction and maybe to reconsider or at least critically discuss the selected situations. The authors show the DOS of pristine PCO in Figure 3. They then show the DOS with the additives and considering adsorbed oxygen in Figure 4. It would be useful to show just the DOS with the additive as an intermediary step.

We agree with the Reviewer that Figure 4 is indeed very crowded and we split the Figure in two. We also substantially expanded the discussion of the oxygen exchange kinetics. Originally, we deliberately kept this discussion short, as many aspects of it are still unclear, but ultimately, we believe that an extended discussion will only encourage readers to consider these results in future investigations of the oxygen exchange reaction.

We extended the discussion part (for a version with all citations, please refer to the revised manuscript file):

Given that the exact mechanism of the oxygen exchange reaction is still not clarified, and many different, partially conflicting theories have been brought forward, it is not possible to present a fully comprehensive theory of the effects of decorations on the oxygen exchange kinetics as of yet. We will, however, discuss the experiments in light of previous results and provide a hypothesis that might reconcile different interpretations. Two important points must be considered in this discussion:

- According to earlier studies, decorations do not alter the oxygen exchange reaction mechanism but accelerate the reaction rate of the equilibrium reaction, encompassing both oxygen reduction and oxygen evolution. Possibilities to alter the equilibrium reaction rate include lowering the kinetic barrier of the rate limiting step, changing participating defect concentrations and altering the energetics of reaction steps, thereby providing a higher

Response to the Reviewers Comments

reactant concentration for the rate limiting step. Several previous studies have further shown that decorations do not alter the apparent activation energy of the oxygen exchange reaction systematically. This activation energy is a complex convolute of contributions from equilibrium constants, defect concentrations and the kinetic barrier of the rate limiting step (and possibly even potential-dependent terms). It is therefore unlikely that a pure decrease of the kinetic barrier of the rate limiting step is the only cause for increased reaction rates.

- Previous studies of surface decorations have presented conflicting interpretations regarding their working mechanism. While originally, the main effect of surface decorations was attributed to changes in the free surface electron concentration, corresponding changes in XPS have not yet been found, and corresponding conductivity changes could not be reproduced on thin film samples. Instead, adsorption thermodynamics have been discussed as the main reason for the observed changes in oxygen exchange kinetics and the results of this study support this theory.

At the core of our hypothesis, surface decorations affect the energy levels of adsorbed reaction intermediates, such as peroxy or superoxy species adsorbed at the surface that are likely dissociating during the oxygen incorporation reaction. At this point it is noteworthy that it is unlikely to observe such molecular oxygen adsorbates directly during NAP-XPS, due to their low concentration and because their signal would be masked by higher concentrated bulk oxygen and SO_4^{2-} adsorbates (which are unavoidable at these conditions). However, as shown in S.I.2, it might indeed be possible to provide evidence for these intermediates with sophisticated XPS measurements.

Here, we suggest that low ionic potential (basic) decorations favor the donation of charge to adsorbed oxygen molecules (as has previously been confirmed for metal surfaces and Na coadsorption) and vice versa for high ionic potential (acidic) decorations. This concept is an alternative view on the original argument of electron concentration changes, however, upon basic decoration, charge does not accumulate in PCO but is rather localized on oxygen adsorbates. This process is schematically shown in Fig. 5d) and leads to different energetics (and concentrations) of charged O_2^- adsorbates on the decorated surfaces. This is further supported by XPS measurements under anodic polarization, where an increased oxygen potential, and hence an increased concentration of surface oxygen species, leads to a work function increase, indicating a more negative surface charge. It is important to note that the band bending picture in this figure is strongly simplified and only illustrates surface dipole formation. In reality, particularly at high temperatures, we expect charge carrier concentrations in the decoration and the surface to evolve according to a complicated interplay of electrostatic and structural effects. This is again supported by the fact that previous studies were generally not able to find corresponding changes in XPS.

Computational results support this hypothesis (see Fig. 5 a-c): the energetic positioning of the 2p orbital of the upper O atom of an O_2 molecule on a surface oxygen site gradually shifts towards lower energies (with regard to E_F) the more basic the surface and the lower the work function becomes. This leads to a higher concentration of molecular adsorbates in equilibrium conditions at high temperatures, which accelerates the oxygen incorporation for basic

Response to the Reviewers Comments

decorations and inhibits it for acidic decorations (we suspect similar processes to be responsible for the acceleration of the oxygen release reaction as well). We also expect that the higher partial charge on molecular oxygen favors dissociation as the O-O bond length increases. As the second main effect, we suggest that defect concentrations on the surface will be substantially altered during decoration. This is also supported by the shifting O 2p band center, which suggests a changing reducibility of the surface. Quantitative estimates, how decorations affect defect concentrations at high temperatures are, however, rather speculative and require a more detailed investigation, possibly including ab-initio thermodynamic approaches. The here presented concept has important implications for studies on the oxygen exchange reaction and poses a further step towards a comprehensive understanding of degradation processes and optimization strategies for solid oxide cell materials.

Regarding DOS plots, we deliberately chose not to display densities of state for decorated surfaces because the structure of these surfaces is simply not known. This encompasses both the atomic arrangement, as well as the actual stoichiometry. Therefore, we believe that putting too much emphasis on these densities of state is not beneficial to the discussion, as they do not allow for deductions about defect concentrations in high temperature. For a molecular oxygen adsorbate, this situation is different, as its DOS captures the effects of the surrounding decoration (which are qualitatively accurate for the present decoration structures), but its own structure is determined by its molecular oxygen character.

To provide more insight into the electronic situation before adsorption, we the PDOS for a Pr atom beneath the decorations without any oxygen adsorbates in the S.I. and, as expected, unoccupied states of Pr shift towards the Fermi energy for basic decorations and vice versa for acidic decorations. However, we strongly want to emphasize that this is not a realistic situation at ambient conditions at high temperatures.

We have added the following section to the S.I.:

S.I. 5. Pr density of states for decorated surfaces

Here, we aim to provide additional information about pristine surfaces after decoration but without the impact of oxygen adsorbates. While an in-depth analysis of surface decoration atoms themselves might be misleading, since both the exact atomic configuration, as well as the precise stoichiometry of the decoration are not known, we investigated a subsurface Pr atom below the decoration. In general, the results are as expected from a surface dipole perspective, with unoccupied Pr levels shifting towards the Fermi level for more basic surfaces. This analysis also showcases that charge redistribution likely not only concerns Pr as the redox active species but also oxygen atoms in the decoration and the subsurface. At this point it is important to emphasize that this analysis is only of limited use for realistic models, since at high temperatures, oxygen vacancy formation in both the decoration and the host material, as well as oxygen adsorbates will lead to a much more complicated charge redistribution between all participants.

Fig. S7. Partial density of states of a Pr atom in the sub-surface for pure PCO and PCO decorated with SrO and SnO₂. In addition, the centroid of the unoccupied Pr4f states is shown, illustrating a shift of the Pr4f level to the Fermi level for more basic surfaces.

The authors have not introduced any additional electrons to the pristine PCO, as a result it is represented as an intrinsic semiconductor. In the depicted schematics, there are no electrons in the conduction band, the acceptor level (adsorbed oxygen) would then have to take electrons from the valence band. As a result, using this model, it would be expected that upon oxygen adsorption PCO, at least near the surface, shows p-type conduction. SrO is expected to insert electrons into the PCO, e.g. would insert electrons into the conduction band (must occur this way as no other free band is available). Alternatively, assuming this model, SnO₂ should extract electrons from the valence band again causing p-type surface near behaviour. At the examined temperatures and oxygen conditions, PCO is known to be redox active.

In fact, the molecular oxygen adsorbate was added at the site of a surface oxygen, as we would expect as a result of adsorption into a surface vacancy. This is now specified in the manuscript. This was the method of choice, as it does not require any kinetic or mechanistic considerations and no additional polaron introduction and just gives a qualitative picture of charge redistribution on the surface. Therefore, we expect that the only effect that is observed in these calculations is the differences in charge redistribution to the oxygen molecule depending on the surface decoration.

We specified the configuration of the added oxygen molecule:

Computational results support this hypothesis (see Fig. 5 a-c): the energetic positioning of the 2p orbital of the upper O atom of an O₂ molecule on a surface oxygen site gradually shifts towards lower energies (with regard to E_F) the more basic the surface and the lower the work function becomes.

Regarding p-type and n-type conductivity, again the discussion is highly complex. In principle, we agree with the Reviewer that in an ideal system without the presence of oxygen, SrO would facilitate reduction of PCO and vice versa for SnO₂ (as is also qualitatively confirmed by the PDOS of Pr – see above). However, at high temperatures, the formation of

Response to the Reviewers Comments

oxygen vacancies in both decoration and bulk, as well as the adsorption of molecular oxygen severely complicates the picture. In addition, we suspect that, in the case of surface decorations, it is (at least partially) surface oxygen that is redox active and that is involved in charge transfer. We are currently trying to develop a model that captures the effects of acidity and basicity on a molecular orbital level, similar to the d-band model of Hammer and Norskov (10.1016/0039-6028(96)80007-0), that might do a better job in explaining a decoration's impact on oxygen exchange (see response of the next comment), but we ask the Reviewer to understand that these are only preliminary ideas that go beyond the scope of this paper and may be discussed in a future manuscript.

This has even been verified by the authors themselves (<https://pubs.rsc.org/en/content/articlelanding/2022/TA/D1TA07128A>). The formation of oxygen vacancies are compensated by the reduction of Pr(IV) to Pr(III). The relatively low mobility of the electronic charge carriers is typically interpreted to indicate small polaron conduction, e.g. electrons in localized Pr4f bands. See paper by JJ Kim Chem. Mater. 2014, 26, 3, 1374–1379 In this case an acceptor would therefore likely remove electrons from the Pr4f band, while a donor would add electrons to it. This situation is more inline with the change in conduction described by Nicollet et al. previously, decrease in conduction as a result of the acidic additive and increased conduction due to the basic additive. If possible, in consideration of this behaviour, could the situation be more accurately modelled? Maybe by using the DOS of Michel et al. with an additional electron (polaron) or by using a slab with defects as the starting point? The Ce4f band should also be considered, as an additional localized level energetically under the conduction band.

This is an important comment and again, we agree with the Reviewer that Pr is usually considered the redox active species and that the common assumption is that electronic charge carriers are more or less localized as small polarons in the Pr impurity band. However, we want to stress two aspects in this discussion. First, again, the reduction of Pr upon basic decoration could not be confirmed experimentally and studies of the absolute conductivity on porous bar samples (as in the original paper of Nicollet et al.) could not be reproduced in a thin film setup with a much better control over geometry and decoration amount (10.1021/acsami.3c03952). In addition, considering the observed work function change and the very small screening length of PCO (in the range of 1 nm, due to the high doping content), only very small amounts of charge have to be redistributed between decoration and host material, which might indeed be simply caused by electron density redistribution between surface and subsurface atoms. This is again in agreement with DFT calculations which see partial charge differences on subsurface atoms but no difference in the central layers of the bulk. Therefore, we believe that a simple electron transfer picture is not particularly suitable in this case, and we expect that the bulk electronic structure of PCO will not be able to account for the experimentally and computationally encountered results on the much more complicated decorated surfaces. In the following we will attempt to present a preliminary picture of the evolution of the electronic structure upon decoration and oxygen adsorption that is based on the d-band model that has been used to describe adsorption on transition metal surfaces.

Response to the Reviewers Comments

The d-band model of Hammer and Norskov relies on the interaction of an adsorbate with a metal d-band and the subsequent creation of bonding and antibonding states in the metal-adsorbate collective. In the present case, the surface is covered by a binary oxide, with electrons primarily present in the O2p band. According to calculations, basic decorations lead to a shallower oxygen 2p band and vice versa (this is also plausible considering the definition of the Smith Acidity). As in the d-band model, we assume that interaction of the O₂ molecule with the O2p band will give rise to bonding and antibonding states. Depending on the relative positioning of the O2p band and the molecular level, the antibonding state will be filled or unfilled. While a higher d-band for transition metals leads to less filling of antibonding states and thus to a stronger binding of adsorbates, we suspect that an analog situation occurs here, with shallower O2p bands leading to less filled antibonding states and stronger adsorption. It is important to note that it is not trivial to deduce the PDOS of adsorbed oxygen from such a picture, as bonding and antibonding states will have different character depending on their vicinity to the adsorbate level or the surface O2p band center.

Again, we want to stress that these are preliminary ideas that might point into the right direction but detailed studies and further computational investigations of adsorbates on decorated, mixed conducting surfaces are necessary to confirm such a model. It seems, however, that experimental and computational results are so far in agreement with this idea.

Minor point: As in order to be removed from the material the electron must cross the surface the vacuum level should not bend but simply shift.

Response to the Reviewers Comments

In Fig. 5 d) we show a very simplified band diagram that visualizes charge redistribution in the system MIEC/decoration+adsorbates/vacuum. First, a decoration induces a surface dipole according to its acidity. This charge redistribution between host material and surface is drawn as band bending near the surface (the spatial extent of charge redistribution is very low, as discussed above) in the left part of the figure. Subsequently, the oxygen adsorbate level is placed according to vacuum level alignment with the already decorated surface. Upon actual adsorption, charge is transferred to the oxygen adsorbate, leading to a superimposed charge redistribution effect, which is again illustrated as (additional) band bending. Instead of band bending, one might also draw a step in the energy diagram to emphasize the localized nature of charge redistribution, but intuitively, we find that more confusing than helpful.

2. Relates to original comment 9: It is still not clear how a lowering of the product energy (dissociated oxygen) would result in an increased rate, assuming that the activation energy and the starting state are the same. This seems like it would change the thermodynamics but not necessarily the kinetics.

We thank the Reviewer for the question and will detail our reasoning. The circumstance that surface decorations alter the equilibrium reaction rate of both incorporation and evolution without altering the apparent activation energy considerably is still mysterious to us and we have not yet found a model that comprehensively (and quantitatively) describes all of these effects. The aspect mentioned by the Reviewer importantly includes energy shifts before and after the rate limiting step. If for example adsorbed molecular oxygen is energetically favorable on basic surfaces (which is strongly indicated by both computational and experimental results), and the rate limiting step is dissociation of molecular oxygen, we expect an increase in reactant concentration for this reaction step and hence an increased reaction rate. At the same time, the kinetic barrier might not be substantially affected, but the overall effect on the apparent activation energy is not clear. At the current stage, we do simply not have enough information to speculate beyond this stage. It would be highly interesting to attempt to isolate concentration contributions to the activation energy (10.1039/D1TA07128A) and investigate concentration amended activation energies for both reaction directions and we hope that we manage to perform these experiments in the near future. If the Reviewer has further input ideas on this topic, we are more than happy to discuss them in more detail (possibly outside the review process).

As we did not put enough emphasis on this explanation, we also added a clarifying text to the discussion on page 6:

Possibilities to alter the equilibrium reaction rate include lowering the kinetic barrier of the rate limiting step, changing participating defect concentrations and altering the energetics of reaction steps before the rate limiting step, **thereby providing a higher reactant concentration for this step.**

3. Minor comment about former 6: The authors should add a bit more information about why the slab is symmetric making their reasoning stronger by citing relevant literature (e.g.

Response to the Reviewers Comments

Dipole correction for surface supercell calculations, Lennart Bengtsson, Phys. Rev. B 59, 12301 – Published 15 May 1999)

We added two references and an additional explanatory sentence:

While dipole correction schemes for asymmetric slabs are available (10.1103/PhysRevB.59.12301, 10.1103/PhysRevB.102.045403), in this study, it is crucial to avoid undesired countercharge effects that might be introduced by nonideal correction.

Previous Response to the First Reviewers Comments

Comments/Response/Changes

Reviewer #1 (Remarks to the Author):

The article, titled "Engineering Surface Dipoles on Mixed Conducting Oxides with Ultra-Thin Oxide Decoration Layers," explores the feasibility of altering the catalytic activity of oxide surfaces through the modification with ultra-thin layers. This modification leads to changes in surface dipoles and work function. Additionally, the authors propose considering the ionic potential of surface cations as a potential descriptor for these effects. Those are interesting and novel results. The paper presents both experimental and computational results, featuring high-quality discussions and can be considered for publication in Nature Communications after revisions.

We thank the Reviewer for their assessment of our manuscript. We have expanded the discussion according to the Reviewer's comments and have also added impedance spectroscopic investigations to the supplementary information (as we wanted to keep the manuscript concise, we chose this approach over adding a new chapter to the manuscript). We also expanded our XPS discussion. Please find our detailed response below.

1. The authors should include additional discussion regarding the rationale for considering ionic potential as a valuable descriptor. They might also explain why it is preferable over parameters such as surface acidity.

We introduced the ionic potential, as it is in our opinion a more fundamental quantity than the Smith Acidity, and, in contrast to the Smith acidity, is a property of the surface cation, and not of a binary oxide. We expect that a real decorated surface deviates from the bulk structure of the binary oxide and that the complex electrostatic situation at the surface of an oxide is indeed strongly dependent on the cations at the surface. In addition, the ionic radius is known for different oxidation states and offers higher flexibility compared to the Smith acidity, which is based on the determination of reaction enthalpies of selected binary oxides.

Response to the Reviewers Comments

We expanded the discussion of surface descriptors in the S.I.:

In general, the comparison shows the expected result that the here explained metrics correlate very well with each other and are similarly well suitable to describe the properties of binary oxides. Differences emerge in their ease of use when considering more complicated oxides and in particular their surfaces. There, the Smith acidity, which is based on empirical data from binary oxides (and which also considers the structure of the products of reactions between binary oxides) is potentially not the best choice to describe the effects induced by modification processes. We suspect that real decorated surfaces deviate considerably from binary oxide structures and that it is the introduced cation that is decisive for the observed changes. Here we suggest that acidity and basicity (which are intuitive concepts and therefore desirable) are better correlated to other underlying ion-specific metrics such as the ionic potential or the electronegativity, which may be better suited to tackle more complicated problems. The ionic potential also has two further advantages: i) ionic radii are listed for different oxidation states, increasing the flexibility of the descriptor, ii) its constituents can be estimated by computational approaches, facilitating the synergy of experimental and theoretical studies. An overview of the quantitative correlations of the discussed metrics is given in the following figure (deviations for the ionic potential of SiO_2 might be due to the strong covalent character of the bond and our use of the formal 4+ charge):

2. More comprehensive details on the electrochemical impedance spectroscopy (EIS) measurements are needed. It would be beneficial if the authors could incorporate EIS data, such as Nyquist plots. The paper mentions that the enhancement in catalytic activity results from improved charge transfer between the surface and absorbed oxygen. Do the EIS results support this conclusion?

We added an additional section in the supporting information on decoration effects and their investigation with impedance spectroscopy. There, we summarize and refer to results published earlier that add more details on the electrochemical methods used. In particular, we added measurement results of impedance spectroscopy during the decoration of LSC64 and PCO10 with SrO and SnO_2 , as these are the relevant material combinations presented in the main manuscript:

S.I.5. In-situ impedance spectroscopy during decoration

For a more in-depth discussion, we want to note here that detailed impedance spectroscopic investigations on thin films upon basic and acidic decoration have been presented by the authors in previous articles^{11,16,17}. Generally, all i-PLD measurements follow a similar procedure, which has been outlined in previous studies¹⁸. The investigated samples consist of a YSZ single crystal electrolyte, Ti/Pt current collecting grids on both sides of the substrate and a 200-300 nm nanoporous LSC64 counterelectrode on one side. On the other side, the working electrode is grown during i-PLD. The temperature during deposition is controlled very precisely via the ohmic offset, which contains the electrolyte resistance (that is well known from literature), as well as resistive contributions from wiring and the current collecting grids (that are measured beforehand). Resulting impedance spectra usually consist

Response to the Reviewers Comments

of three major contributions. i) the above mentioned ohmic offset, ii) a mid-frequency semicircle, which corresponds to the surface exchange resistance coupled with the chemical capacitance of the working electrode, and iii) a low-frequency arc, which corresponds to the surface exchange resistance coupled with the chemical capacitance of the counter electrode. The volume-related chemical capacitance of the counter electrode is much higher due to the higher thickness, leading to different characteristic frequencies of the two electrode contributions, and thus to well separable semicircles. It is worth mentioning, that for some samples, a small high-frequency shoulder appears, which is attributed to interfacial resistances between thin film and electrolyte.

As the working electrode feature is usually a nearly perfect semicircle, it is very unlikely that diffusion limitations affect the measurements (they would be visible as Warburg-type distortions on the high frequency side of the semicircle) and thus, the observed resistance is directly connected to the rate of the rate determining step of the oxygen exchange reaction. Our previous results have suggested that this rate determining step is related to charge transfer and dissociation¹⁹, but the details about this reaction mechanism are still not entirely clear. It is, however, very likely, that upstream reaction steps of the oxygen reduction reaction include adsorption of molecular oxygen and charge transfer onto the adsorbed oxygen.

During decoration, the only impedance contribution that is affected is the working electrode feature. Upon basic decoration, the surface exchange resistance decreases, suggesting faster oxygen exchange kinetics and vice versa for acidic decoration. In the figure below, exemplary impedance measurements of acidic and basic decoration for LSC64 and PCO are shown.

Fig. S7. a) Impedance spectra of $\text{La}_{0.6}\text{Sr}_{0.4}\text{CoO}_{3-\delta}$ in its pristine state as well as with 2 pls SrO and SnO_2 decoration. b) Impedance spectra of $\text{Pr}_{0.1}\text{Ce}_{0.9}\text{O}_{2-\delta}$ in its pristine state as well as with 2 pls SrO and SnO_2 decoration.

3. The O1s spectra display a main peak at approximately 528 eV (see Fig. S2). However, this peak appears to lack perfect symmetry. It would be interesting to know if the authors observed any high-energy shoulder on the peak, similar to what has been reported in the literature (e.g., see DOI: 10.1002/adfm.202108005). A thorough analysis of the O peak could aid in identifying the concentration of defective oxygen or oxygen vacancy concentration (<https://doi.org/10.1002/er.4613>).

Response to the Reviewers Comments

We absolutely agree, that the O1s main peak might include information about oxygen lattice defects, and we considered this possibility in our interpretation. However, the situation here is slightly more complicated, as we deal with a perovskite with a metallic electronic structure. In fact, it is likely that this asymmetry is caused by the metallicity of LSC, as is also strongly indicated in the results of a previous study (10.1021/acs.jpcc.5b08596), where the asymmetry increases from $\text{SrTi}_{1-x}\text{Fe}_x\text{O}_{3-\delta}$ to $\text{La}_{1-x}\text{Sr}_x\text{FeO}_{3-\delta}$ to $\text{La}_{1-x}\text{Sr}_x\text{CoO}_{3-\delta}$. Hence, a lack of the asymmetry is observed for perovskites with a similar vacancy concentration but a lower conductivity and a less metallic electronic structure. In addition, the asymmetry does not change considerably when increasing the oxygen partial pressure during near-ambient-pressure XPS, as would be expected from the changing oxygen vacancy content.

We adapted the corresponding passage in the S.I.:

The work function is highest on SnO_2 decorated LSC and lowest for SrO decorated LSC. The main oxygen 1s species exhibits some asymmetry which is attributed to the metal-like electronic structure of LSC (it has previously been shown that this peak asymmetry correlates with the metallicity of the electronic structure of perovskite oxides¹²).

Reviewer #2 (Remarks to the Author):

This seems to be an highly interesting paper, but I realize that I do not have the expertise to evaluate the manuscript only after I have agreed on reviewing it. Specifically, I am not in the field of thin-film oxides for SOFC, nor do I perform DFT calculations. Basically, I am not familiar with the status quo, hence, am not in the position to evaluate the novelty of the work. However, I do have a couple of points for the authors to explain, perhaps, in their revision.

(1) The concept of ionic potential, is averaged over all cationic sites on a surface, why is that so? does this mean that the anions on the surface are completely irrelevant as far as properties such as surface dipole and oxygen exchange are concerned? how would one treat host materials with different levels of anionic vacancies then?

(2) Other than the XPS-based measurements of work function, is there any more characterizations? UPS for example to confirm the work function result, STM/STS for atomically resolved structure of adsorbates, or scanning Kelvin probe for combined imaging and work function measurements? Lack of structural characterization makes the conclusions a little unsettling.

We thank the Reviewer for their careful evaluation of our manuscript, for their honesty regarding their review, and for their very interesting comments and remarks. We will discuss the two main concerns below.

- 1) The definition of the ionic potential we used in this study is the ratio of ionic charge and ionic radius. One convenient property of this descriptor is the fact that the ionic radii of cations are listed not only for different valence states but also for different coordination numbers. Thereby, different surface structures can be described with this

Response to the Reviewers Comments

metric. To account for more complex chemistries, we chose to average over the surface cationic sites as a first approximation. Hence, it allows to tentatively account for the differences between varying surface compositions, e.g. with a Sr-rich termination being more basic than a mixed surface with both, AO and BO₂ terminations. Regarding anion vacancies, on the one hand changing cation valences could account for the charge redistribution caused by these vacancies. On the other hand, we suspect that the surface vacancy concentration is actually related to the cation ionic potential, as it reflects the binding strength between oxygen and surface cations. We agree, however, that this metric is not able to quantify acidity or basicity changes due to stoichiometry changes on one and the same surface (e.g. caused by oxygen partial pressure changes). We rather see the metric as a means for a versatile comparison between different surface chemistries.

We added the following passage to reflect this discussion in the manuscript (page 3):

It is worth mentioning, that averaging over the surface cation stoichiometry is only an estimate allowing for the comparison of different surface chemistries and does not account for the oxygen vacancy concentration at the surface as a function of temperature and oxygen partial pressure.

- 2) We agree with the Reviewer that we paid too little attention to a convincing structural characterization in the first version of the manuscript. At the same time, we ask the Reviewer to understand that it is not a trivial task to analyze the outermost surface chemistry of complex oxides at elevated temperatures. Before we detail the changes that we made to the manuscript, we will briefly discuss the different techniques mentioned by the Reviewer. It is true that UPS is a more common way to measure work functions, because the energy scale calibration with a Fermi edge kinetic energy of ~20 eV is more precise than that of an XPS spectrometer. However, the UPS spectrometer that is available to us is not compatible with sample heating and contacting options that are required for these experiments. Also, the photon source itself does not influence the sharpness or position of the secondary electron cutoff energy, as these secondary electrons always undergo many inelastic interactions. Consequently, work function differences found between the studied samples are equally accurate with XPS and UPS. However, the absolute values rely on a calibration standard (111-oriented Pt thin film). STM/STS and in particular the in-situ version of this technique (connected to the PLD chamber) would be the ideal tool to precisely characterize the studied surfaces and their evolution upon decoration. However, the characterization of thin film surfaces on the atomic scale with STM is best done on ideal surfaces such as the SrTiO₃ or La_{0.8}Sr_{0.2}MnO_{3-δ} surfaces that were studied in the group of Prof. Ulrike Diebold at TU Wien. The thin films studied in this work are by far not as ideal (as can also be seen in the AFM images that we added to the supporting information). Upon request at an earlier point during this study, Prof. Diebold predicted that it would take a substantial amount of time (at least several months) to reach a point of thin film quality, where she would expect publishable atomic scale STM results. We see this work as a proof of concept and we are convinced

Response to the Reviewers Comments

that in the future, atomic scale characterization combined with a fitting computational approach has strong potential to unveil even more details behind these complex but very powerful surface modifications.

The additional characterization that was accessible to us was atomic force microscopy and more low energy ion scattering (LEIS) measurements, and we added the corresponding new results of LSC and PCO surfaces to the supporting information. The AFM images show no differences between decorated and undecorated surfaces and hence, we can confidently exclude the possibility that the decoration agglomerates in the form of surface particles. While LEIS results of decorated PCO thin films suggest nearly full coverage with the decoration, for LSC, the host material cations are still visible on the surface, however, a stronger decoration ion peak appears, making for 56 % of the total surface cation signal (for a nominal decoration thickness of one monolayer). Based on these results, we conclude that the decoration grows as a very flat layer on the surfaces, eventually leading to a full coverage of the surfaces, but possibly not necessarily in layer-by-layer growth. We are aware that the models adopted in DFT calculations represent a very idealized system (also causing significantly stronger work function trends as observed experimentally) but considering the combination of all experimental and computational results, we are convinced that we successfully describe the fundamental properties of the investigated system.

The following section was added to the supporting information:

S. 7. Extended sample characterization

In Fig. S9, atomic force microscopy (AFM) images of LSC and PCO surfaces with different decorations are shown. These measurements were performed for previous publications¹¹ and showcase that the decorations do not lead to any particle formation on the surface, or in fact to any visible alteration of the surface at all. For LSC, polycrystalline thin films with SnO₂ and CaO decorations were investigated. In both cases, the granular surface of the LSC thin film is completely unchanged and no visible traces of the decoration (nominally one unit cell) can be seen on the surface. In the case of PCO, the thin films were deposited on a YSZ/GDC system, leading to epitaxial thin film growth, with atomic terraces being visible in AFM. Again, both SnO₂ and CaO decorations do not agglomerate, and the very flat surface remains unchanged during decoration (for SnO₂ decoration, LEIS measurements found a coverage of >90% of the surface cations with Sn, see below). Some few isolated particles are found on the surfaces, which are attributed to dirt on the sample surface.

Response to the Reviewers Comments

Fig. S9. AFM images of LSC and PCO surfaces upon decoration with SnO₂ and CaO. In both cases, no visible traces of the decoration can be seen in AFM images.

In addition to AFM measurements, low energy ion scattering (LEIS) measurements yield further insight into the chemistry of the outermost surface of PCO and LSC thin films. In a previous publication¹¹, LEIS was used to investigate PCO decorated with SnO₂ and SrO. For one nominal unit cell of SrO, no Ce or Pr signal could be identified during LEIS measurements, for SnO₂, the Sn signal amounts to around 90 % of the signal, indicating a high degree of coverage. For LSC, a SnO₂ (1 ML) decorated sample was investigated with a 5 keV ²⁰Ne⁺ primary analysis beam to allow for a proper separation of different cation signals. Again, the Sn signal is the largest signal contribution and dominates the surface cation chemistry (56 % of the total cation signal), however, the coverage does not seem to be as complete as for the case of PCO. Considering the low amounts of deposited material and the unchanged morphology during AFM measurements, the experimental evidence conclusively indicates that the decorations grow very flat and ultimately lead to full coverage, but they do not necessarily grow in a layer-by-layer growth mode. In the case of PCO, also depth profiles were recorded with LEIS¹¹, showing a sharp decrease of the decoration ion signal below the surface, showing no significant interdiffusion into the host material.

Response to the Reviewers Comments

To combine chemical analysis with lateral resolution, we also performed a secondary ion mass spectrometry measurement on a $100 \times 100 \mu\text{m}^2$ area of SnO_2 decorated LSC. The measurement was performed in the collimated burst alignment (CBA) mode (10.1039/C3JA50059D) which allows for high lateral resolution. Again, an overlay of the Sn and the Sr signal (probing depth around 1 nm) reveals no noticeable agglomeration in larger particles or islands.

The combination of these results emphasizes the complexity of these surfaces, particularly at high temperatures, but also showcases that the decoration layers prefer a flat growth mode vs. island or particle growth. This also is reflected in the XPS results, which probe the average over the whole surface. The fact that XPS trends are not as pronounced as the corresponding DFT model calculations (in particular for LSC) might indicate that we indeed do not deal with a perfectly homogeneous monolayer coverage but with a somewhat nonideal coverage, i.e. only the majority of the surface is covered small amounts of the host material remain exposed at the surface.

Reviewer #3 (Remarks to the Author):

The authors present an excellent study on the effect of ultra-thin oxide layers on the surface of mixed-conductors. Interestingly they proposed a new descriptor, the ionic potential, to describe how the work function and the surface exchange kinetics can be altered depending on the decoration used. I think this work provided valuable insights into surface behaviour

Response to the Reviewers Comments

and will be of great interest to those working in the field of mixed-conductors as well as the broader scientific community. The experiments and simulations appear to have been meticulously carried out, the manuscript is well-written, and the data is well-presented. I've carefully read through the manuscript a couple of times, and I did not find anything that I believe needs to be changed. As such, I recommend this work to be published as-is. I would like to congratulate the authors on an excellent and very interesting study.

We thank the Reviewer for their assessment of our manuscript and are delighted that they find the results interesting and insightful.

Reviewer #4 (Remarks to the Author):

The paper is an extension of the acidity basicity work previously presented by Nicollet et al. The unique ability of the authors to control the additive layer is interesting and the idea of interfacial dipole formation is intriguing. More information, however, and, in particular deeper contextualization/discussion of the findings is needed to make the manuscript publishable.

We thank the Reviewer for their careful evaluation of our manuscript and will discuss the comments point by point below. We are grateful for this review, as it constructively challenged our work with very valid arguments and gave us the opportunity for a thorough improvement of our manuscript with additional data and revised discussions, which ultimately led to a better explanation of our experimental and computational results.

- 1) What is meant by strongly acidic and highly basic. This should be nuanced as SrO is less "basic" than e.g. Li₂O and SnO₂ is considered less "acidic" than e.g. WO₃. It should be more clearly justified why these two decorations were specifically selected?

This assignment was based on the Smith acidity scale, and we agree that it is beneficial to put the terms "strongly basic" and "strongly acidic" into perspective. We adjusted the text correspondingly. These oxides were chosen, as they span the broadest acidity range while still being relatively easy to work with and to deposit with PLD. More acidic oxides, such as MoO₃ or WO₃, might contaminate the PLD chamber and using more basic oxides, such as BaO was attempted, but these were unfortunately not stable during target synthesis. We added this justification to the manuscript.

We provided additional information about the oxides in the introduction (page 1):

We probed the effects of SrO, a highly basic oxide (-9.4 on the Smith acidity scale, similar basicity as Li₂O at -9.2), and SnO₂, a strongly acidic oxide (2.2 on the Smith acidity scale, being slightly less acidic than WO₃ at 4.7), on the work function and the electronic structure of La_{0.6}Sr_{0.4}CoO_{3-δ} (LSC) and Pr_{0.1}Ce_{0.9}O_{2-δ} (PCO), two chemically and structurally different MIEC oxides.

Response to the Reviewers Comments

We expanded the sample preparation section (page 7):

SnO₂ and SrO as decoration materials were chosen to span a broad range of acidity, while being manageable in terms of preparation (strong hydroxide and carbonate formation in air for more basic oxides, such as BaO) and contamination (volatile compounds for more acidic oxides, such as WO₃).

- 2) Although the authors assert that the surface of the metal oxides has been decorated with a monolayer of either SnO₂ or SrO based on results previously attained on PCO, no morphological information is provided in the manuscript. It is unclear if there is verification for LSC. Ideally LEIS (the authors argue this method is suitable) for the samples shown in Fig.1 should be provided. Evidence is needed that a monolayer is formed and that no island/particle formation occurs.

We fully agree with the Reviewer that we did not discuss the sample morphology satisfyingly in the original manuscript. We didn't expect it to be advisable to perform LEIS on the same samples that were measured in XPS, as several temperature and voltage treatments were performed on these samples, and these treatments will likely have altered the surface chemistry considerably. We therefore performed further LEIS measurements on LSC decorated with SnO₂, which we added to a new section (S.7. Extended sample characterization) in the supporting information, together with atomic force microscopy images of decorated LSC and PCO. The LEIS spectra on LSC still show some signals of the host materials cations, which are overshadowed by a strong Sn signal. The same sample was also measured by AFM to investigate possible particle or island formation. Since AFM results clearly show that the morphology is completely unaltered, we can exclude the agglomeration to particles or distinct islands on the surface. We see with LEIS, however, that the coverage is not 100%, and it seems that it is less than for the same decoration amount on PCO. Considering that the Sn signal still amounts to 56% of the total cation signal in LEIS measurements, we conclude that the decoration layer forms in a very flat manner, ultimately leading to a full coverage (as seen for PCO) but not necessarily in a layer-by-layer growth. We believe that this is also reflected by our XPS results, which show weaker trends than are predicted by DFT calculations. As XPS takes an area-weighted average over local work functions for a larger spot size (10.1103/PhysRevApplied.19.037001), the possibility of a not fully covering decoration layer must be considered in the interpretation of our results. For further experimental evidence, we also performed secondary ion mass spectrometry measurements on a SnO₂ decorated LSC thin film in collimated burst alignment (CBA) mode, which allows for relatively high spatial resolution. On a 100 x 100 μm² scan, no signs of island formation or spatial inhomogeneity could be detected. We concretized the manuscript in critical passages and paid more attention to an accurate description of our surfaces.

We adapted the text on page 2:

As low energy ion scattering (LEIS) measurements and atomic force microscopy images show very high surface coverage for PCO and LSC thin films decorated with SrO and SnO₂¹⁴ as well as no signs of island or particle growth (see S.7 in the supporting information), we assume

Response to the Reviewers Comments

a flat growth mode of the decoration layer which ultimately leads to a homogenous coverage. Computational results as well as i-PLD experiments support this assumption (see below). While XPS measurements probe a certain sample volume and cannot distinguish between island growth and full layer coverage, LEIS truly probes the outermost surface chemistry and is thus an invaluable tool to investigate modified surfaces.

We added a discussion about the link between computational and experimental results on page 5:

At this point, we also want to link computational results to the real surface morphology. Based on our surface analysis with LEIS and AFM the decoration layers grow very flat and do not form islands or particles, but possibly also do not cover the surface fully for thin decoration layers (see S.7. in the supporting information). Comparing experimental work function measurements with DFT predictions, we observed that the experimentally found trend is indeed weaker than predicted by DFT (especially for LSC, which seems to exhibit a higher fraction of free surface). As XPS work function measurements take an area-weighted average over the work function in a certain spot size²³ and thus might also record signal from small amounts of undecorated surface, this is in good agreement with the surface morphologies and chemistries of our samples.

The following section was added to the supporting information:

S. 7. Extended sample characterization

In Fig. S9, atomic force microscopy (AFM) images of LSC and PCO surfaces with different decorations are shown. These measurements were performed for previous publications¹¹ and showcase that the decorations do not lead to any particle formation on the surface, or in fact to any visible alteration of the surface at all. For LSC, polycrystalline thin films with SnO₂ and CaO decorations were investigated. In both cases, the granular surface of the LSC thin film is completely unchanged and no visible traces of the decoration (nominally one unit cell) can be seen on the surface. In the case of PCO, the thin films were deposited on a YSZ/GDC system, leading to epitaxial thin film growth, with atomic terraces being visible in AFM. Again, both SnO₂ and CaO decorations do not agglomerate, and the very flat surface remains unchanged during decoration (for SnO₂ decoration, LEIS measurements found a coverage of >90% of the surface cations with Sn, see below). Some few isolated particles are found on the surfaces, which are attributed to dirt on the sample surface.

Response to the Reviewers Comments

Fig. S9. AFM images of LSC and PCO surfaces upon decoration with SnO_2 and CaO . In both cases, no visible traces of the decoration can be seen in AFM images.

In addition to AFM measurements, low energy ion scattering (LEIS) measurements yield further insight into the chemistry of the outermost surface of PCO and LSC thin films. In a previous publication¹¹, LEIS was used to investigate PCO decorated with SnO_2 and SrO . For one nominal unit cell of SrO , no Ce or Pr signal could be identified during LEIS measurements, for SnO_2 , the Sn signal amounts to around 90 % of the signal, indicating a high degree of coverage. For LSC, a SnO_2 (1 ML) decorated sample was investigated with a 5 keV $^{20}\text{Ne}^+$ primary analysis beam to allow for a proper separation of different cation signals. Again, the Sn signal is the largest signal contribution and dominates the surface cation chemistry (56 % of the total cation signal), however, the coverage does not seem to be as complete as for the case of PCO. Considering the low amounts of deposited material and the unchanged morphology during AFM measurements, the experimental evidence conclusively indicates that the decorations grow very flat and ultimately lead to full coverage, but they do not necessarily grow in a layer-by-layer growth mode. In the case of PCO, also depth profiles were recorded with LEIS¹¹, showing a sharp decrease of the decoration ion signal below the surface, showing no significant interdiffusion into the host material.

Response to the Reviewers Comments

To combine chemical analysis with lateral resolution, we also performed a secondary ion mass spectrometry measurement on a $100 \times 100 \mu\text{m}^2$ area of SnO_2 decorated LSC. The measurement was performed in the collimated burst alignment (CBA) mode (10.1039/C3JA50059D) which allows for high lateral resolution. Again, an overlay of the Sn and the Sr signal (probing depth around 1 nm) reveals no noticeable agglomeration in larger particles or islands.

The combination of these results emphasizes the complexity of these surfaces, particularly at high temperatures, but also showcases that the decoration layers prefer a flat growth mode vs. island or particle growth. This also is reflected in the XPS results, which probe the average over the whole surface. The fact that XPS trends are not as pronounced as the corresponding DFT model calculations (in particular for LSC) might indicate that we indeed do not deal with a perfectly homogeneous monolayer coverage but with a somewhat nonideal coverage, i.e. only the majority of the surface is covered small amounts of the host material remain exposed at the surface.

- 3) The authors argue that SnO_2 is more acidic due to its high 4+ charge and small ionic radius while Sr is basic as it is only +2 and has a large radius. In addition to the more common SnO_2 , SnO has been known to form, i.e. Sn would only have a +2 charge. As this is a key point, evidence is needed for the oxidation state of Sn, e.g. information possible from XPS? As a reference: work that discusses the transition of thin films

Response to the Reviewers Comments

from SnO to SnO₂ and or Sn the authors should refer to [10.1016/j.heliyon.2016.e00112](https://doi.org/10.1016/j.heliyon.2016.e00112). Rationale for why XPS instead of the more common UPS was used to determine the work function. <https://doi.org/10.1016/j.apsusc.2009.11.002>

Considering that the kinetic effect of SnO_x decoration on the oxygen exchange reaction and on the work function is in good agreement with its acidity according to the Smith acidity scale, we deem it very unlikely that it is actually present as SnO. In addition, Sn⁴⁺ is the most stable oxidation state for Sn and preferred in the predominant number of compounds. We can of course not exclude the possibility of mixed oxidation states of Sn on the surface, and consequentially also the possibility of deviation from the perfect SnO₂ stoichiometry. We revisited our XPS measurements, and a single Sn 3d 5/2 peak appears at 485.8 eV, which is at the lower end for SnO₂ and the upper end for SnO. Ultimately, computational and experimental results point towards a rather acidic SnO_x decoration, but again, we want to emphasize, that for a quantitative and thorough understanding of the electronic effects induced by such ultrathin decorations, precise atomic-scale surface characterization and corresponding computational models are necessary.

We extended our discussion on page 4:

At this point, we want to emphasize that, similarly to our computational models, also the actual stoichiometry of the deposited decoration layer is unclear (and hardly accessible for analytic tools due to the tiny sample volume). While SnO₂ decoration leads to a strong decrease of oxygen exchange kinetics, as would be predicted by its Smith acidity, we cannot exclude the possibility of mixed oxidation states or deviation from ideal oxygen stoichiometry (i.e. SnO_{2-δ}).

Regarding the choice of XPS vs. UPS measurements, we mainly chose XPS because the UPS spectrometer that is available to us is not compatible with sample heating and contacting options that are required for these experiments. Also, the photon source itself does not influence the sharpness or position of the secondary electron cutoff energy, as these secondary electrons always undergo many inelastic interactions. Consequently, work function differences found between the studied samples are equally accurate with XPS and UPS. However, the absolute values rely on a calibration standard (111-oriented Pt thin film).

- 4) Although covered in a previous paper, the authors should mention where the sulfur contamination is coming from during the XPS measurements. Is there a strong affinity for the entire LSC surface to S-species or do the authors believe that specifically e.g. the La-sites are affected?

Generally, we observe that LSC is much more susceptible towards sulphate formation than e.g. PCO, which aligns also with LSC surfaces being more basic than PCO surfaces. Since LSC has been experimentally shown to be largely SrO terminated, we believe that sulphate groups predominantly form with Sr cations. This is also supported by Sr- and S-rich particles that are found on top of the LSC surface after long-term exposure to lab air. Regarding the

Response to the Reviewers Comments

origin of S during XPS measurements, we added a corresponding text passage at the respective location on page 3:

Upon exposure of pristine surfaces to considerable gas phase pressures ($>10^{-3}$ mbar), a new oxygen species appears at 532 eV, alongside a weak corresponding S species, that is caused by the inherent impurity content of measurement atmospheres despite using nominally high purity gases²³.

- 5) What is meant by “chemically suitable configurations” were selected for the modelling?

This phrasing was indeed unfortunate. We adjusted the respective text passage on page 3:

As a proof of concept, we selected geometric configurations for SrO and SnO₂ layers on LSC and PCO based on appropriate bond distances and orientation with regard to the host lattice as a starting point for structural relaxation (for more details refer to S.8. in the supporting information).

- 6) Were the decorations were added to both the top and the bottom of the slab for the calculation (images in supplementary information)? If so why?

Yes, the decorations were added to both the top and the bottom of the slab for the calculations to avoid an overall dipole in the supercell and eventual inaccuracies in the calculation of the work function. This is explained in the Methods section on page 8:

“A symmetric slab was chosen to avoid overall dipole effects which might interact with surface decorations.”

- 7) The authors should elaborate what is meant by, “SrO-terminated LSC and SnO₂ terminated PCO show no significant tendency towards oxygen vacancy or peroxide formation - indicating that the proposed configuration likely reflects an energetically favorable structure.” The authors also state that the formation of one surface vacancy is favorable for SnO₂ terminated LSC. Did the authors do energetic calculations for the adsorption of oxygen on the presented slabs? Were activation energies considered?

We added a more precise explanation to the corresponding text passage on page 3:

SrO-terminated LSC and SnO₂ terminated PCO show no significant tendency towards oxygen vacancy or peroxide formation, i.e. removing a neutral oxygen atom or adding a second oxygen atom to a surface oxygen (and thus forming a peroxide) does not lead to total energy gains - indicating that the proposed configuration likely reflects an energetically favorable structure.

Regarding energetic calculations for adsorbed oxygen, we did not attempt any barrier calculations or reaction step simulations in this study. In a previous study, we did see

Response to the Reviewers Comments

substantial changes of idealized barriers for the adsorption of molecular O_2 into a surface vacancy upon SO_3 adsorbate formation (although the absolute values are not reliable, an increase of the barrier is very likely), so we would not be surprised to see a change in adsorption barriers on decorated surfaces as well due to surface charge formation. However, since the exact adsorption process on these decorated surfaces is unclear, we doubt the overall reliability of such calculations. We revisited the calculations we performed for the densities of state of molecular oxygen on PCO and compared the total energies to the ideal slab and molecular O_2 . In agreement with our interpretation, the total energy decreases by 0.08 eV (more favorable) for O_2 on SrO-decorated PCO and increases by 0.01 eV (more unfavorable) for SnO_2 decorated PCO, confirming that basic decorations favor the presence of O_2 adsorbates. However, we believe that for a true energy diagram, a thorough mechanistic analysis with the calculation of different reaction pathways and scenarios would be necessary, which goes far beyond the scope of this study. In addition, reliable energy evaluations would require knowledge of the energetically favorable stoichiometry, which again would require much more detailed insight into the atomic scale structure of the surface decorations.

8) Calculating the density of states for transition metal oxides is not trivial. Furthermore, the description of heavier elements, such as the lower transition metals and lanthanides, typically involves relativistic effects, such as spin-orbit coupling. Specifically the small polaron conduction of Pr-doped Ceria is usually not correctly captured. The authors cite the work of Michel et al. [10.1021/acs.jpcc.0c05352](https://doi.org/10.1021/acs.jpcc.0c05352) when discussing the use of Hubbard U correction. The DOS presented in Fig. 3f seems to have been done with the inclusion of spin-orbit coupling: This should be clarified!

We fully agree with the Reviewer, and we performed the calculations carefully and in consideration of previous literature results. The choice of the U values relied on literature in this case and in the case of PCO, our calculations were indeed guided by the paper the Reviewer is referring to. Our DOS of PCO, which was evaluated from a spin-polarized calculation with GGA+U without spin-orbit coupling is in very good agreement with the ones the authors present in the mentioned paper ([10.1021/acs.jpcc.0c05352](https://doi.org/10.1021/acs.jpcc.0c05352)) with PCO showing semiconducting behavior with Pr4f intragap states and an O2p-dominated valence band. In addition, as we do not introduce additional electrons (e.g. by O bulk nonstoichiometry), we believe that our results properly reflect the electronic structure of PCO.

We added the following text passage to the methods section:

Since PCO has not been the subject of many computational studies, the resulting density of states was compared with literature and is in very good agreement with previous reports, even though no spin-orbit coupling was used in this approach⁴⁴.

8) The concepts of a surface dipole should be more clearly described. It could be useful to refer to the large body of work from PV on layered oxide structures for how best to approach such an explanation. A discussion similar in detail to that presented in the paper of Klein et al. ([10.3390/ma3114892](https://doi.org/10.3390/ma3114892)) would help the reader understand the

Response to the Reviewers Comments

suggested model. The authors specifically discuss how different sputtering treatments of SnO₂ films can influence surface dipoles. They also report that in some cases surface dipoles can be modified by post-deposition treatments (a point also briefly mentioned in the manuscript). This point goes back to the consideration of decorations that have multiple oxidation states (e.g. Sn). Do the authors have any insights into how robust the descriptor of ionic potential of surface cations is in systems with potentially varying oxidation states? This could be a particularly important consideration for the presented thin films as many of the surface metal sites are likely not stoichiometric. The authors may have made an effort to address this point in Figure 3, but the caption, "Next to SnO₂-decorated LSC (5.72 eV), the work functions of SnO_{1.75} (5.55 eV) and SnO (5.53 eV) decorations are shown additionally, as well as the work functions of SrO_{1.5} and SrO₂ decorations for SrO-decorated PCO." is difficult to understand!

We thank the Reviewer for this comment and have expanded our discussion of the surface dipole. Indeed, the concept of surface dipoles has been widely discussed in different contexts, in particular with regard to surface reduction/oxidation, as suggested by the Reviewer, and also with regard to the formation of surface adsorbates in contact with a given atmosphere. We attempted a more comprehensive description of the surface dipole, which, in this case, is even more complicated as it also entails the heterointerface between host material and decoration. We want to emphasize, that this study doesn't aim for a structurally exact representation of the systems, as this would require atomic-scale knowledge of these surfaces, which requires a much more sophisticated surface analysis as is available to us at this moment. However, we are confident, that this study reveals the fundamental qualitative interactions underlying surface decorations on mixed conducting oxides and will be the basis for future studies unveiling the exact details of these processes.

We expanded the discussion on page 5:

Both mechanisms contribute to the buildup of electric fields on the surface, consequently leading to changes in the work function. At this point, it is important to emphasize, that this analysis was done for idealized, stoichiometric monolayers, showcasing fundamental principles of surface dipole formation. As we deal with mixed ionic and electronic conducting materials at high temperatures and in contact with the gas phase, real interfaces likely exhibit higher complexity.

From an electrochemical point of view, dipole emergence is caused by different electrochemical potentials of electrons in the decoration and the material. Specifically, in the exemplary case of a negative surface charge, the electrochemical potential of the electron initially lies higher than for the decoration. For mixed conductors, depending on a balance between changed defect formation energies (which are also encoded in the electronic structure, e.g. via the O-2p band center²¹) and electrostatic potential buildup, complicated space charge zones might form at high temperatures, entailing altered oxygen vacancy concentrations, nonstoichiometric decoration layers, and complicated reconstructions. Also intermixing between decoration and host material cannot be excluded after longer periods at high temperature. Partly, this is already indicated by our analysis of varying oxygen content

Response to the Reviewers Comments

in decorations shown in Fig. 3 a), where changing oxygen stoichiometry also leads to changes in the work function (examples of the interplay of stoichiometry and surface dipole have also been previously discussed in literature²²). In addition, the formation of adsorbates on a decoration in contact with a gas phase will also affect the surface dipole as well as the space charge layer and induce further charge transfer. An extended discussion of the electrochemical potential landscape for the example of SrO decoration on LSC is given in S.4. of the supporting information.

We also adjusted the mentioned caption:

a) Ab-initio work functions of LSC and PCO surfaces, pure or modified with SrO, SnO₂ and SO₃. In the cluster of values for SnO₂-decorated LSC (5.72 eV), also the work functions of SnO_{1.75} (5.55 eV) and SnO (5.53 eV) decorations are shown additionally.

- 9) From all the studied materials, LSF, STF, LSC and PCO, only LSC and PCO were studied in detail. A rationale for why these two samples were selected should be given. The conclusion of the authors that it is unlikely that the decorations result in changed activation energy while asserting that it is likely that the energetics of the adsorbates are being influenced needs clarification. The authors argue that the surface potential will change the energy levels of adsorbed reaction intermediates, (e.g. peroxo or superoxol). Would this, however, not likely result in a change of the activation energy?

LSC and PCO were studied because they represent two fundamentally different material classes, perovskites and fluorites, and share no common cations. Both exhibit fast oxygen exchange kinetics (in particular in the case of LSC) and both accommodate substantial oxygen vacancy concentrations. Furthermore, i-PLD experiments have shown that various perovskite materials exhibit similar oxygen partial pressure and temperature dependences of their oxygen exchange kinetics, emphasizing the overall similarity of their pristine surfaces. We provided more details about our reasoning in the revised version:

We probed the effects of SrO, a highly basic oxide (-9.4 on the Smith acidity scale, similar basicity as Li₂O at -9.2), and SnO₂, a strongly acidic oxide (2.2 on the Smith acidity scale, being slightly less acidic than WO₃ at 4.7), on the work function and the electronic structure of La_{0.6}Sr_{0.4}CoO_{3-δ} (LSC) and Pr_{0.1}Ce_{0.9}O_{2-δ} (PCO), mixed conducting representatives of two fundamentally different material classes, perovskite and fluorite oxides.

Regarding the second part of this question, the answer is more complicated. Analyzing the available literature on surface infiltration and decoration based on the acidity/basicity concept (which has mostly been done by Nicollet et al., Seo et al. and the authors), the studies seem to agree that the activation energy remains largely unaffected by the surface modification. It is in fact, an unresolved question, how a decoration can change the oxygen exchange kinetics without significantly altering its activation energy. One possibility in our opinion is the proposed idea that the decoration lowers the energy level of specific reaction intermediates. If for example, the rate determining step of the reaction would be the dissociation of charged

Response to the Reviewers Comments

molecular oxygen adsorbates, and a basic decoration would favor the presence of these adsorbates energetically, it would be possible that the kinetic barrier for the rate determining step remains approximately the same. This idea seems also to be supported by our computational analysis. Furthermore, we see this theory substantiated by the fact that experimental evidence for peroxo-species (as described in the next comment/answer) could only be found for LSC with a basic surface decoration. We want to stress, that the discussion about the activation energy and the kinetic impact of surface decorations on the oxygen exchange reaction is still unresolved and we believe that the experimental results we present in this study will help to build a better understanding of this problem.

Minor point out of curiosity: The authors state that it is unlikely to observe molecular adsorbates directly during near ambient pressure XPS, due to their low concentration and because their signal would be masked by higher concentrated bulk oxygen and SO_2^{-4} adsorbates (unavoidable). Previously in the manuscript the authors argue that peroxo-species are likely present on the surface and visible in the XPS. What about the peroxo-species would likely make them visible in that particular case?

We thank the Reviewer for this remark, we did not communicate our findings clearly enough. During standard near ambient pressure XPS, peroxo-species will not be visible due to two circumstances. On the one hand, oxygen partial pressures during NAP-XPS are limited to the mbar range. On the other hand, unavoidable SO_4 adsorbates cause an oxygen signal at the same energy where we expect peroxo-species. During this study, we performed the first attempts of a new measurement approach during XPS. We performed experiments on SrO decorated LSC in UHV and at relatively low temperature (450 °C), such that oxygen exchange is strongly inhibited. By applying a controlled amount of bias voltage vs. an Fe/FeO oxygen reservoir on the counter electrode side, we were able to increase the effective oxygen partial pressure in the working electrode beyond the mbar range and to flood the surface with oxygen. Thereby, we could substantially increase the concentration of the desired peroxo-species at the LSC surface without the presence of acidic impurity species and investigate the effect on the oxygen XPS signal. Here, we suspect that peroxo-species cause an O1s species at around 532 eV, whose intensity can be tuned by adjusting the bias voltage and thus the species concentration on the surface.

We added a more detailed explanation to chapter S.I.2:

Critical to this approach is the fact that measurement are performed in UHV and also at relatively low temperatures (450 °C) Thereby, the surface exchange is very slow and anodic polarization leads to a very high oxygen chemical potential at the working electrode surface, facilitating the formation of these peroxide adsorbates. In addition, the XPS signature is visible in UHV, because no SO_4^{2-} adsorbates are present on the surface in these conditions.

REVIEWERS' COMMENTS

Reviewer #4 (Remarks to the Author):

The authors have addressed the comments.